# Fractional-statistics-induced entanglement from Andreev-like tunneling

Gu Zhang [1,2,3], Pierre Glidic [4,5], Frédéric Pierre [4], Igor Gornyi [6,7] & Yuval Gefen [8]

The role of anyonic statistics stands as a cornerstone in the landscape of topological quantum techniques. While recent years have brought forth encouraging and persuasive strides in detecting anyons, a significant facet remains unexplored, especially in view of connecting anyonic physics to quantum information platforms—whether and how entanglement can be generated by anyonic braiding. Here, we demonstrate that even when two anyonic subsystems (represented by anyonic beams) are connected only by electron tunneling, entanglement between them, manifesting fractional statistics, is generated. To demonstrate this physics, we rely on a platform where fractional quantum Hall edges are bridged by a quantum point contact that allows only transmission of fermions (so-called Andreev-like tunneling). This invokes the physics of two-beam collisions in an anyonic Hong-Ou-Mandel collider, accompanied by a process that we dub anyon-quasihole braiding. We define an entanglement pointer—a current-noise-based function tailored to quantify entanglement associated with quasiparticle fractional statistics. Our work, which exposes, both in theory and in experiment, entanglement associated with anyonic statistics and braiding, prospectively paves the way to the exploration of entanglement induced by non-Abelian statistics.

One of the most fascinating classes of quasiparticles is known as anyons. Recent years have borne witness to an intensified spotlight on anyons within the condensed-matter community. The focal point of this scrutiny stems from the fact that anyons exhibit fractional statistics, which touches on the very foundations of quantum mechanics. Furthermore, anyons may represent a promising toolbox for quantum information processing (see, e.g., refs. 1,2). These quasiparticles, defying conventional exchange statistics, are predicted to reside in topologically intricate states, e.g., those realized in the regime of fractional quantum Hall (FQH) effect[3,4]. In particular, anyonic quasiparticles are hosted by the edges of Laughlin quantum-Hall states. The landscape of anyons extends to encompass Majorana modes, foreseen to materialize at the

edges of topological superconducting materials[5,6]. Three decades have passed since the pioneering confirmation of the fractional charge of Laughlin quasiparticles[7,8]. Inspired by earlier endeavors in the exploration of fractional statistics (see e.g., refs. 9–11), highly persuasive signals of anyonic statistics have been directly and indirectly observed in Fabry-Perot[12–15] and Hong-Ou-Mandel interferometers[16–19].

This leap in the search for anyonic statistics has been accompanied by a series of landmark experiments that have unveiled a plethora of exotic anyonic features in FQH systems. Among these are the existence of charge neutral modes[20,21], fractional Josephson relation[22], and Andreev-like tunneling[17,23–27] in anyonic systems[28]. The agreement between the experimental findings and the theoretical predictions not

[1]National Laboratory of Solid State Microstructures, School of Physics, Jiangsu Physical Science Research Center, Nanjing University, Nanjing 210093, China. [2]Collaborative Innovation Center of Advanced Microstructures, Nanjing University, Nanjing 210093, China. [3]Beijing Academy of Quantum Information Sciences, Beijing 100193, China. [4]Université Paris-Saclay, CNRS, Centre de Nanosciences et de Nanotechnologies, 91120 Palaiseau, France. [5]NanoLund and Solid State Physics, Lund University, Box 118, 22100 Lund, Sweden. [6]Institute for Quantum Materials and Technologies, Karlsruhe Institute of Technology, 76131 Karlsruhe, Germany. [7]Institut für Theorie der Kondensierten Materie, Karlsruhe Institute of Technology, 76131 Karlsruhe, Germany. [8]Department of Condensed Matter Physics, Weizmann Institute of Science, Rehovot 761001, Israel. ✉e-mail: igor.gornyi@kit.edu

only consolidates our understanding but also offers new horizons to fuse the physics of anyons with other foundational themes of quantum mechanics. Indeed, in addition to earlier theoretical ideas[29–40], recently, there has been another surge of theoretical proposals[41–50] on understanding and detecting anyonic features, and possibly harnessing them for quantum information processing platforms (see, e.g., refs. [1,2,51,52]).

Entanglement is another fundamental quantum-mechanical element and a prerequisite for the development of quantum technology platforms. Despite its significance, experimentally quantifying entanglement remains a challenging endeavor. Recently, ref. [53] proposed to measure entanglement stemming from quantum statistics of quasiparticles by a certain combination of the current cross-correlation functions. The main message of that reference, addressing integer quantum Hall platforms, is that the statistics-induced entanglement targets genuine entanglement (manifest via collisions between indistinguishable quantum particles), without resorting (cf. refs. [54–56]) to the explicit study of Bell's inequalities[57] and, thus, establishes a possibility of directly accessing entanglement in transport experiments. Notably, extracting statistics-induced entanglement is far from being a trivial task, as statistical properties need not necessarily lead to entanglement, which is, for instance, the case for a fermionic product state.

When transitioning to anyonic systems, the quantification of entanglement becomes even more formidable. This is, in particular, related to the lack of readily available fractional statistics in "natural platforms," which hinders the development of intuition about the statistics-induced mechanisms of entanglement (like bunching and antibunching for bosons and fermions, respectively). Furthermore, the quasiparticle collisions that can directly reveal anyons' statistics[36] through entanglement are now commonly believed to be irrelevant to the noise measurements in anyonic Hong-Ou-Mandel colliders[16–18]. Indeed, when considering dilute anyonic beams, anyonic collisions are overshadowed by time-domain braiding[39–42] of an incoming anyon with spontaneously generated quasiparticle-quasihole pairs. Nevertheless, measurements of anyons' entanglement through their collisions hold immense potential for characterizing and manipulating anyonic states. Despite the importance of anyonic statistics, the quest to generate, observe, and quantify anyonic statistics-induced entanglement remains a challenge to this day.

Here, we take on this challenge and investigate, both theoretically and experimentally, entanglement generated among particles of two subsystems, which is underlain by anyonic statistics. We employ the so-called "Andreev-like tunneling" platform, where the central collider allows only fermions to tunnel between anyonic edge channels. Since the braiding phase of anyons with fermions is trivial, this setup does not support time-domain braiding at the collider. We demonstrate that, nevertheless, anyonic statistics affects the correlations between the two subsystems. This is the result of braiding between a dilute-beam anyon and a quasihole, the latter being triggered by an Andreev tunneling event. We refer to this process as an anyon-quasihole braiding, and demonstrate that it gives rise to the dependence of collision-induced entanglement on the fractional braiding phase.

## Results

### Entanglement pointer for Andreev-like tunneling

In this work, we combine anyonic statistics with quantum entanglement and define the entanglement pointer to quantify the statistics-induced entanglement in a Hong-Ou-Mandel interferometer on FQH edges with filling factor $\nu$ (Fig. 1a). Our platform contains three quantum point contacts (QPCs), including two diluters (Fig. 1b) and one central collider (Fig. 1c). These QPCs bridge chiral channels propagating at different sample edges (indicated by red and blue arrows of Fig. 1a). The setup is characterized by the experimentally measurable transmission probabilities $\mathcal{T}_A$, $\mathcal{T}_B$, and $\mathcal{T}_C$ of the two diluters and the central QPC, respectively.

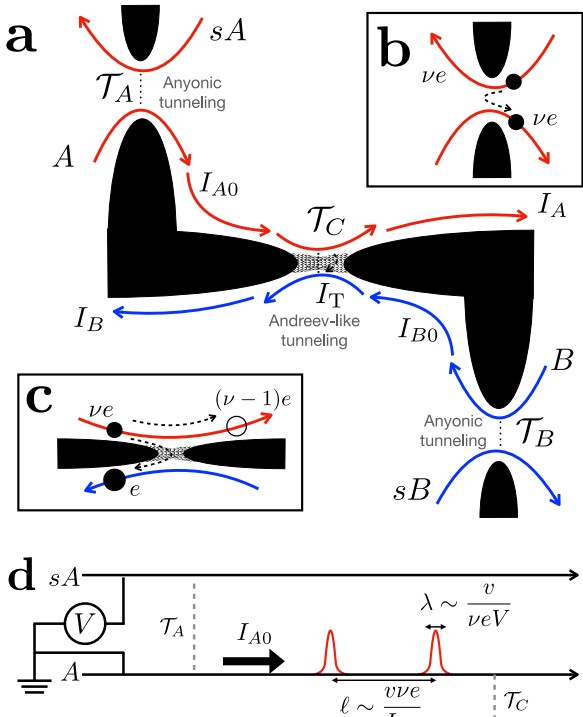

**Fig. 1 | Schematic depiction of the model with Andreev-like tunneling between the fractional edges (cf. Fig. S6 in the Supplementary Information).** The quantum Hall bulk is represented by white regions separated by potential barriers ("fingers" introduced by gates) shown in black; the gray areas correspond to barriers allowing for electron (but not anyon) tunneling. **a** The entire setup involves two source arms ($sA$, $sB$) and two middle arms ($A$, $B$) in the FQH regime. They host chiral anyons that correspond to the bulk filling factor $\nu < 1/2$. Chiral edge-state transport modes are designated with the red and blue curved arrows for subsystems $\mathcal{A}$ (including $sA$ and $A$) and $\mathcal{B}$ ($sB$ and $B$), respectively. $I_A$ and $I_B$ represent the currents in middle arms ($A$ and $B$, respectively), past the central QPC. Before the central QPC, currents in arms $A$ and $B$ are represented by $I_{A0}$ and $I_{B0}$, respectively. Current $I_T$ tunnels through the central QPC connecting arms $A$ and $B$. **b** Anyons of charge $\nu e$ tunnel from the sources to corresponding middle arms $A$ and $B$ through diluters with transmissions $\mathcal{T}_A$ and $\mathcal{T}_B$, respectively. **c** Channels $A$ and $B$ communicate through the central QPC (central collider) with the transmission $\mathcal{T}_C$. The central QPC allows only electrons to tunnel, resulting in the "reflection" of an anyonic hole [with charge $(\nu - 1)e$, empty circle], which resembles Andreev reflection at the metal-superconductor interfaces. **d** Theoretical depiction of subsystem $\mathcal{A}$ that comprises channels $sA$ and $A$ (the upper half of **a**). Channel $A$ features the dilute current beam $I_{A0}$, coming from source $sA$ through a diluter with transmission probability $\mathcal{T}_A$. The schematics of subsystem $\mathcal{B}$ are similar. Channel $A$ is strongly out of equilibrium, when the typical distance between neighboring non-equilibrium anyons $l$ is much larger than their typical width $\lambda$.

As far as the two diluter QPCs are concerned, they are characterized by a small rate of anyon tunneling through, generating dilute non-equilibrium beams in channels $A$ and $B$. These beams are characterized by two length scales (see Fig. 1d): (i) the typical width of non-equilibrium anyon pulses, $\lambda \sim v/\nu e V$, and (ii) the typical distance between two neighboring non-equilibrium anyon pulses, $\ell \sim v\nu e/I_{A0}$ (and similarly for channel $B$), where $v$ is the velocity of edge excitations. When $I_{A0} \ll (\nu e^2)V$, we obtain $\lambda \ll \ell$, such that incoming anyons can typically be considered as well-separated and, thus, independent quasiparticles. This is the regime of diluted anyonic beams addressed here, characterized by the weak-tunneling condition for diluters, $\mathcal{T}_{A,B} \ll 1$. For later convenience, we define $I_+ = I_{A0} + I_{B0}$ as the sum of non-equilibrium currents $I_{A0}$ and $I_{B0}$ in arm $A$ and $B$ (see Fig. 1a), respectively, before arriving at the central collider.

The model at hand is crucially distinct from more conventional anyonic colliders[16–18,39,41] in that its central QPC only allows the

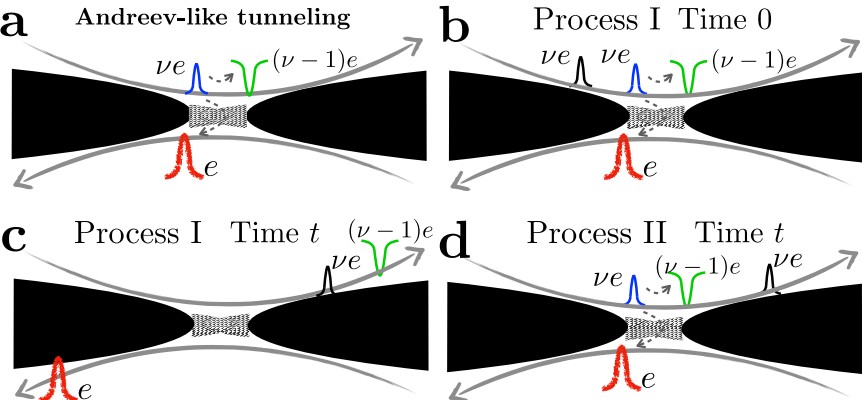

**Fig. 2 | Anyon-quasihole braiding in Andreev-like tunneling processes.** Gray arrows mark the chirality of the corresponding channels. **a** Leading-order Andreev-like tunneling, corresponding to Fig. 1c. Here, an anyon from the diluted beam (the blue pulse, of charge $\nu e$) arrives at the central collider, triggering the tunneling of an electron (the red pulse, of charge $e$) and the accompanied reflection of a hole [the green pulse, of charge $(\nu - 1)e$]. Panels b and c: Illustrations of Process I, at times 0 and $t$, respectively. In (**b**), an Andreev-like tunneling occurs at time 0 at the collider. The positions of the particles involved at a later time $t$ are marked accordingly in (**c**). In both (**b**) and (**c**), the non-equilibrium anyon (the black pulse)

is located upstream (to the left) of the reflected hole (the green pulse). **d** Process II, in which the Andreev-like tunneling occurs at time $t$, when the black pulse has already passed the central QPC. In comparison to (b) and (c), here the non-equilibrium anyon (the black pulse) is located downstream (to the right) of the reflected hole (the green pulse). The interference of Processes I and II thus generates the anyon-quasihole braiding between the non-equilibrium anyon (black) and the reflected quasihole (green). Note that this is not a vacuum-bubble braiding (ref. 40) a.k.a. time-domain braiding (refs. 18,39,41,42,71).

transmission of fermions[25,27,28]. This is experimentally realizable by electrostatically tuning the central QPC into the "vacuum" state (no FQH liquid), thus forbidding the existence and tunneling of anyons inside this QPC. The dilute non-equilibrium currents in the middle arms are carried by anyons with charge $\nu e$ (Fig. 1b), where $\nu$ is the filling fraction. Since only electrons are allowed to tunnel across the central QPC, such a tunneling event must be accompanied by leaving behind a fractional hole of charge $-(1 - \nu)e$; the latter continues to travel along the original middle edge (Fig. 1c). This "reflection" event is reminiscent of the reflection of a hole in an orthodox Andreev tunneling from a normal metal to a superconductor; hence, such an event is commonly dubbed "quasiparticle Andreev reflection"[24,50]. As distinct from the conventional normal metal-superconductor case, in an anyonic Andreev-like tunneling process, (i) both the incoming anyon and reflected "hole" carry fractional charges, and (ii) the absolute values of anyonic and hole charges differ.

It is known that for anyonic tunneling, time-domain braiding (or, alternatively, braiding with the topological vacuum bubbles[40]) can occur between an anyon-hole pair generated at the central QPC and anyons that bypass the collider[39–42]. Such a process is, however, absent for Andreev-like tunneling, where vacuum bubbles are made of fermions that cannot braid with anyons. Instead, another mechanism of braiding is operative in Andreev-like setups, which requires the inclusion of higher-order tunneling processes at the diluters. As shown in Fig. 2 (where we take the single-source case as an example), the fractional statistics of a fractional-charge hole (the green pulse in Fig. 2), which is left behind by the fermion tunneling, enables braiding of this quasihole with anyons supplied by the diluter (black pulses of Fig. 2). We term this type of braiding "anyon-quasihole braiding". For comparison, time-domain braiding in an anyonic tunneling system is illustrated in Section II in the Supplementary Information (SI).

Anyon-quasihole braiding significantly influences the generation of entanglement between the two parts of the system—subsystems $\mathcal{A}$ and $\mathcal{B}$ (see Fig. 1a). To characterize this statistics-induced entanglement we introduce the entanglement pointer (cf. ref. 53) for Andreev-like tunneling:

$$\mathcal{P}_{\text{Andreev}} \equiv -S_{\text{T}}(\mathcal{T}_A, \mathcal{T}_B) + S_{\text{T}}(\mathcal{T}_A, 0) + S_{\text{T}}(0, \mathcal{T}_B). \quad (1)$$

Here, $S_{\text{T}}(\mathcal{T}_A, \mathcal{T}_B)$ refers to the noise of the tunneling current between the two subsystems, which is a function of the transmission probabilities of the two diluters. The entanglement pointer effectively subtracts out those contributions to the tunneling noise that are present when only one of the two sources is biased (which is equivalent to setting one of the two transmissions to zero), thus highlighting the effects of correlations formed between the two diluted anyonic beams. Indeed, the contribution to the noise that results from collisions between particles from the two beams is absent in the sum of two single-source processes; the corresponding difference is captured in Eq. (1). By construction, $\mathcal{P}_{\text{Andreev}}$ naturally quantifies entanglement generated by two-particle collisions (see Fig. 3 for the illustration of corresponding correlations), through which quasiparticle statistics is manifest (in analogy with bunching and antibunching for bosons and fermions, respectively). Although the entanglement pointer is defined relying on the tunneling-current noise, it can also be measured with the cross-correlation noise, see Eq. (5) below and the ensuing discussion.

**Tunneling-current noise**

As discussed above, anyonic statistics is manifest in Andreev-like tunneling through the central collider and the corresponding tunneling-current noise. Remarkably, although the tunneling particles are fermions, the associated transport still reveals the anyonic nature of the edge quasiparticles. This is the consequence of anyon-quasihole braiding between reflected fractional-charge qusihole (green pulse in Fig. 2) and an anyon from diluted beams (black pulse in Fig. 2, generated by tunneling through diluters). More specifically, for Andreev-like transmission through the central QPC at $\nu < 1/2$, the expression for the tunneling noise, when the two sources are biased by the same voltage $V$, can be decomposed as follows: $S_{\text{T}} = S_{\text{T}}^{\text{single}} + S_{\text{T}}^{\text{collision}}$, where (for simplicity, here and in what follows, we set $\hbar = \nu = 1$)

$$S_{\text{T}}^{\text{single}} = \text{Re}\left\{ \mathcal{T}_C\,\mathcal{T}_A \frac{\nu e^3 V}{2} \frac{(2\pi\nu)^{1-\nu_s} e^{i\pi(\nu_s - \nu\nu_s + 1)}}{\pi\nu \sin(\pi\nu_s) + 2f_1(\nu)\mathcal{T}_A} [2i\pi\nu - \mathcal{T}_A(1 - e^{-2i\pi\nu})]^{\nu_s - 1} \right\} + \{A \to B\},$$

$$\text{with } f_1(\nu) \equiv (\nu_s - 1)\sin(\pi\nu)\{\sin[\pi(\nu_s - \nu)] + \sin(\pi\nu)\},$$

$$(2)$$

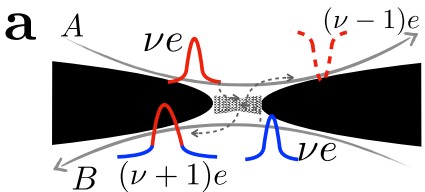
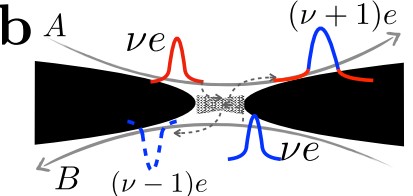

**Fig. 3 | Generation of cross-correlation by the collision of two anyons (the red and blue pulses) arriving at the collider from uncorrelated sources.** There are two possible processes. **a** A pulse of charge $(\nu+1)e$ is generated in channel $B$, leaving an outgoing fractional-charge hole with charge $(\nu-1)e$ in channel $A$. **b** An alternative process, where a pulse of charge $(\nu+1)e$ is created in channel $A$, leaving a quasihole with charge $(\nu-1)e$ in Channel $B$. The resulting cross-correlation is intrinsically related to the entanglement generated between two subsystems ($\mathcal{A}$ and $\mathcal{B}$), which is captured by the entanglement pointer. The Andreev two-anyon collision processes are further "decorated" by anyon-quasihole braiding (which involves additional anyons supplied by the diluters, cf. Fig. 2). To keep the figure simple, the latter is not shown.

is the sum of single-source noises resulting from separately activating sources $sA$ and $sB$, and

$$S_T^{\text{collision}} = \text{Re}\left\{ \mathcal{T}_C\, e^3 V \frac{\sqrt{\mathcal{T}_A \mathcal{T}_B}\, f_2(\nu)\cos(\pi\nu_d/2)}{\pi\nu\,\sin(\pi\nu_s) + 2f_1(\nu)\sqrt{\mathcal{T}_A\mathcal{T}_B}} \left[ \mathcal{T}_A(1 - e^{-2i\pi\nu}) + \mathcal{T}_B(1 - e^{2i\pi\nu}) \right]^{\nu_d - 1} \right\},$$

$$\text{with } f_2(\nu) \equiv \frac{4\pi^3 (2\pi\nu)^{1-\nu_d}\, \Gamma(1-\nu_d)}{\sin(2\pi\nu)\,\Gamma(1-2\nu)\,\Gamma(1-\nu_s)},$$
$$(3)$$

is the double-source "collision contribution" (see Section IIIB of the SI). In Eqs. (2) and (3), $\nu_s \equiv 2/\nu + 2\nu - 2$ and $\nu_d \equiv 2/\nu + 4\nu - 4$ reflect the scaling features of Andreev-like tunneling, for the single-source and the double-source (collision-induced) contributions, respectively. Crucially, a phase factor $\exp(\pm 2i\pi\nu)$ appearing in Eqs. (2) and (3) is generated by the braiding of two Laughlin quasiparticles (i.e., by the anyon-quasihole braiding) that have the statistical phase $\pi\nu$. This factor, multiplying the diluter transparency, is, however, concealed in the single-source noise in the strongly dilute limit ($\mathcal{T}_{A,B} \ll 1$) by the constant term $-2i\pi\nu$ in the square brackets of Eq. (2). On the contrary, this statistical factor appears already in the leading term of Eq. (3), rendering the collision contribution to the noise particularly handy for extracting the information on the quasiparticle statistics.

According to Eq. (1), the entanglement pointer is determined by the value of $S_T^{\text{collision}}$:

$$\mathcal{P}_{\text{Andreev}} = -S_T^{\text{collision}}. \qquad (4)$$

Note that $S_T^{\text{collision}}$ vanishes when $\nu=1$, indicating that the Andreev entanglement pointer, $\mathcal{P}_{\text{Andreev}}$, represents a quantity that is unique for anyons. As an important piece of the message, when $\nu=1/3$, the extra noise induced by collisions between two Laughlin quasiparticles is negative, i.e., $S_T^{\text{collision}}(\mathcal{T}_A, \mathcal{T}_B) < 0$. This indicates that the simultaneous arrival of anyons reduces the probability of Andreev-like reflection at the central collider, as supported by the experimental data (cf. Fig. 4).

The entanglement pointer, $\mathcal{P}_{\text{Andreev}}$, has three advantages over the total tunneling-current noise $S_T(\mathcal{T}_A, \mathcal{T}_B)$ or current cross-correlations. Firstly, it rids of single-beam contributions to the current correlations, which are not a manifestation of genuine statistics-induced entanglement. Secondly, $\mathcal{P}_{\text{Andreev}}$ reflects the statistics-induced extra Andreev-like tunneling for two-anyon collisions. It provides an alternative option (other than the braiding phase[12] and two-particle bunching or anti-bunching preferences[36]) to disclose anyonic statistics. Thirdly, $\mathcal{P}_{\text{Andreev}}$ is resilient against intra-edge interactions, in edges that host multiple edge channels. In the setup we consider here, the interaction occurs between the edges coupled by the central QPC. Since the region where the two edges come close to each other has a rather small spatial extension, the effect of such inter-edge interaction is weak, leading to small corrections to both cross-correlation and the entanglement pointer (see SI Section VB). The situation will be, however, different, when considering systems with complex edges that contain multiple edge channels. Indeed, following our discussions at the end of

Section VC in the SI, interactions in such setups may lead to a significant correction to the cross-correlation due to the so-called charge fractionalization. This correction, which may even exceed the interaction-free noise, is avoided by the subtraction of the single-source noises, when evaluating the entanglement pointer.

## Physical interpretation of the entanglement pointer

The essence of an entanglement pointer can be illustrated by resorting to single-particle (Fig. 2) and two-particle (Fig. 3) scattering formalism revealing the statistical properties of anyons in the course of two-particle collisions, via bunching or anti-bunching preferences[36,58]. The situation is more involved for the model under consideration, as particles that are allowed to tunnel at the central collider (fermions) are of distinct statistics that differs from that of the colliding particles (anyons). As we have emphasized above, although only fermions can tunnel through the central collider, anyonic statistics still manifests itself by influencing the probability of Andreev-like tunneling events, when two anyons arrive at the collider simultaneously.

The probability of a two-anyon scattering event is proportional to $\mathcal{T}_A \mathcal{T}_B$; Andreev-like tunneling then produces fractional charges on both arms, as shown in Fig. 3. These processes establish the entanglement between two subsystems $\mathcal{A}$ and $\mathcal{B}$, that are initially independent from each other otherwise. Noteworthily, here, the entanglement is induced by the statistics of colliding anyons, not interactions at the collider. After including both single-particle and two-particle scattering events, we obtain (SI Section VI) the differential noises at a given voltage $V$:

$$s_T = (s_T)_{\text{single}} + (s_T)_{\text{collision}} = (\mathcal{T}_A + \mathcal{T}_B)\mathcal{T}_C - \left(\mathcal{T}_A^2 + \mathcal{T}_B^2\right)\mathcal{T}_C^2 + \mathcal{T}_A \mathcal{T}_B P_{\text{Andreev}}^{\text{stat}},$$

$$s_{AB} = (s_{AB})_{\text{single}} + (s_{AB})_{\text{collision}}$$

$$= -(1-\nu)\mathcal{T}_C(\mathcal{T}_A + \mathcal{T}_B) - \mathcal{T}_C(\nu - \mathcal{T}_C)\left(\mathcal{T}_A^2 + \mathcal{T}_B^2\right) - \mathcal{T}_A \mathcal{T}_B P_{\text{Andreev}}^{\text{stat}},$$
$$(5)$$

where the subscripts "single" and "collision" indicate contributions from single-particle and two-particle scattering events, respectively. Here, $s_T = \partial_{eI_+/2} S_T$ and $s_{AB} = \partial_{eI_+/2} S_{AB}$ are the differential noises, and $S_{AB} = \int dt \langle \delta\hat{I}_A(t)\delta\hat{I}_B(0)\rangle$ is the irreducible zero-frequency cross-correlation with $\delta\hat{I}_{A,B} \equiv \hat{I}_{A,B} - I_{A,B}$ the fluctuation of the current operator $\hat{I}_{A,B}$. The factor $P_{\text{Andreev}}^{\text{stat}}$ refers to extra Andreev-like tunneling induced by anyonic statistics in the course of two-anyon collisions. It would be equal to zero if anyons from subsystem $\mathcal{A}$ were distinguishable from those in $\mathcal{B}$. In this case, the noise would be equal to the sum of two single-source ones. By comparing with Eqs. (2), (3), and (4), $P_{\text{Andreev}}^{\text{stat}}$ can be expressed via the microscopic parameters [see Eq. (S100) of the SI Section VI and more details in SI Secs. I and IV]; furthermore, $\mathcal{T}_{A,B} = \partial_V I_{A0,B0} h/(e^2\nu)$ are directly related to the conductance of the corresponding diluter. As another feature of Andreev-like tunnelings, $S_T$ in Eq. (5) does not explicitly depend on $\nu$, since the central QPC allows only charge $e$ particles to tunnel.

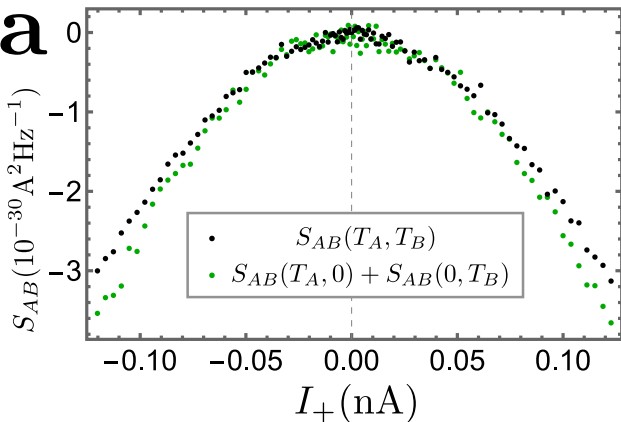

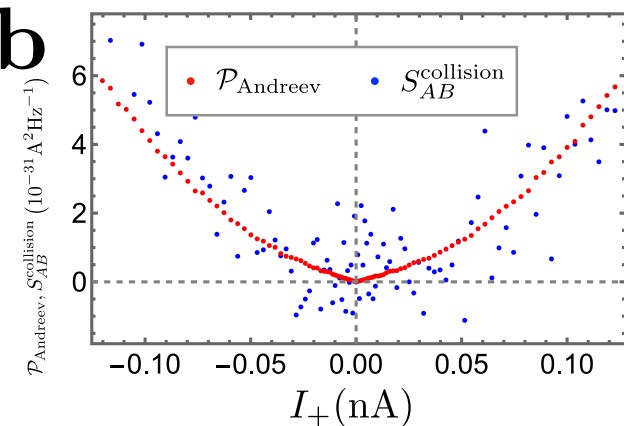

**Fig. 4 | Cross-correlation noise: theory vs. experiment. a** Experimental data for the cross-correlation $S_{AB}$, for the single-source (green points, corresponding to the sum of two single-source signals) and double-source (black points) scenarios, respectively. Here, the x-axis represents $I_+$ of the double-source situation. **b** The theory-experiment comparison. The experimental data $\mathcal{P}_{Andreev}$ refers to the collision contribution to the cross-correlation, obtained following the definition of Eq. (1). For comparison, one can alternatively obtain $S_{AB}^{collision}$, following Section VIII of the SI, indirectly from transmissions at diluters and the central collider. The two approaches compare well for most values of $I_+$. For small values of $I_+$, the weight of thermal fluctuations becomes more significant, leading to a larger deviation. Further experimental information, concerning the transmission probability through the central collider ($\mathcal{T}_C$) and that through the diluters ($\mathcal{T}_A$ and $\mathcal{T}_B$) is provided in Fig. 5 and S7 of the SI, respectively.

Equation (5) exhibits several features of Andreev-like tunneling in an anyonic model. Firstly, in the strongly dilute limit, $s_{AB} \approx (\nu - 1)s_T$, when considering only the leading contributions to the noise, i.e., the terms linear in both $\mathcal{T}_A$ (or $\mathcal{T}_B$) and $\mathcal{T}_C$. Both $(s_T)_{single}$ and $(s_{AB})_{single}$ correspond to $S_T^{single}$ [Eq. (2)] and are subtracted out following our definition of the entanglement pointer, Eq. (1). In both functions, the double-source collision contributions, i.e., the bilinear terms $\pm \mathcal{T}_A \mathcal{T}_B P_{Andreev}^{stat}$, involve $P_{Andreev}^{stat}$. This is the contribution to the entanglement pointer generated by statistics, when two anyons collide at the central QPC. Most importantly, bilinear terms $\propto \mathcal{T}_A \mathcal{T}_B$ of both functions in Eq. (5) have the same magnitude, i.e., $(s_T)_{collision} = -(s_{AB})_{collision}$. Consequently, the experimental measurement of $\mathcal{P}_{Andreev}$, though defined with tunneling current noise, can be performed by measuring the cross-correlation of currents in the drains, which is more easily accessible in real experiments:

$$\mathcal{P}_{Andreev} = -\frac{e\mathcal{T}_A\mathcal{T}_B}{2}\int dI_+ \, P_{Andreev}^{stat}(I_+) = S_{AB}(\mathcal{T}_A, \mathcal{T}_B) - S_{AB}(\mathcal{T}_A, 0) - S_{AB}(0, \mathcal{T}_B).$$

(6)

**Comparison with experiment**

We now compare our theoretical predictions with the experimental data (cf. refs. 27 and [59]), see Fig. 4. Panel a shows the raw data for the double-source noise $S_{AB}(\mathcal{T}_A, \mathcal{T}_B)$ and for the sum of single-source noises $S_{AB}(\mathcal{T}_A, 0) + S_{AB}(0, \mathcal{T}_B)$. For the single-source data, the x-axis represents $I_{A0}(\mathcal{T}_A, 0) + I_{B0}(0, \mathcal{T}_B)$, i.e., the sum of non-equilibrium currents in the two single-source settings (with either source sA or source sB biased). Firstly, as shown in panel a, the double-source cross-correlation, $S_{AB}(\mathcal{T}_A, \mathcal{T}_B)$, is smaller in magnitude compared to the sum, $S_{AB}(\mathcal{T}_A, 0) + S_{AB}(0, \mathcal{T}_B)$, of two single-source cross-correlations. This fact agrees with the negativity of $S_T^{collision}$ (tunneling-current noise induced by two-anyon collision) of Eq. (3) for $\nu = 1/3$. To verify our theoretical result, we compare, in Fig. 4b, the values of $\mathcal{P}_{andreev}$ and $S_{AB}^{collision}$. Here, the former is obtained directly from the measured noises by virtue of Eq. (6), while the latter, defined as $S_{AB}^{collision}(\mathcal{T}_A, \mathcal{T}_B) \equiv S_{AB}(\mathcal{T}_A, \mathcal{T}_B) - S_{AB}(0, \mathcal{T}_B) - S_{AB}(\mathcal{T}_A, 0)$, is calculated from the measured dependence of the tunneling current on the incoming currents using the following relation:

$$S_{AB}^{collision} = \frac{eI_+\tan(\pi\nu)}{2(\nu_d-1)\tan(\pi\nu_d/2)}\left\{\left(\frac{\partial}{\partial I_{A0}}-\frac{\partial}{\partial I_{B0}}\right)[I_T(\mathcal{T}_A,0)+I_T(0,\mathcal{T}_B)-I_T(\mathcal{T}_A,\mathcal{T}_B)]\right\}\bigg|_{I_-=0}.$$

(7)

Derivation of Eq. (7) relies on explicit expressions for the noise, Eqs. (2) and (3), as well as expressions (11) for the tunneling currents presented in Methods (see details in Section VIII of the SI). Figure 4b demonstrates remarkable agreement between the theory and experiment for $\mathcal{P}_{Andreev}$. This indicates the validity of the qualitative picture based on the phenomenon of anyon-quasihole braiding, which influences Andreev-like tunneling as described in the section entitled "Physical interpretation of the entanglement pointer".

## Discussion

In this work, we have studied, both theoretically and experimentally, the generation of entanglement associated with the fractional quasiparticle statistics in an anyonic (with filling factor $\nu < 1/2$) Hong-Ou-Mandel interferometer that exhibits Andreev-like tunneling through the central QPC. We defined the entanglement pointer through the associated noise functions that are obtained by considering anyon-triggered fermion tunneling at the central QPC, which is accompanied by "anyon-quasihole braiding" of Andreev-reflected anyonic charges with anyons from non-equilibrium beams. The Andreev-like tunneling in an anyonic collider is "halfway" between the integer case of ref. 53 (where both tunneling and dynamics along the arms are fermionic) and a purely anyonic collider (both tunneling and dynamics are anyonic). The latter case will be addressed elsewhere, with insights from the present work indicating that quasiparticle collisions do matter in the collider geometry. The identification of the peculiar braiding mechanism in the Andreev-like platform studied here suggests a variety of potential unexpected phenomena determined by anyonic statistics.

The Hong-Ou-Mandel setup in the Andreev-like tunneling regime provides us with a convenient platform for a direct inspection and study of real anyonic collisions, and, especially, the resulting generation of entanglement of initially unentangled anyons in transport experiments (cf. Fig. 3). Our analysis indicates that the exchange of electrons (carrying "trivial" fermionic statistics) between two anyonic subsystems suffices to render those subsystems aware of their mutual anyonic statistics, generating non-trivial statistical entanglement between the subsystems. The theory predictions are verified in the experiment; the measured data agree remarkably well with the theoretically calculated one, for both the current cross-correlations $S_{AB}$ and the entanglement pointer $\mathcal{P}_{Andreev}$. We have thus demonstrated the crucial role of

two-particle collisions in establishing fractional-statistics-induced entanglement in anyonic colliders.

An interesting issue here concerns the role of electrostatic interactions: To what extent is our entanglement pointer sensitive to such interactions? One identifies two types of interactions: intra-edge interactions among several chiral modes in a given edge and inter-edge interactions around the collider. To leading order, the contribution of the former to the current-current correlations is subtracted when calculating the entanglement pointer. The magnitude of the latter is parametrically small, given the relatively small size of the collider and typically weak interaction between the edges (cf. ref. 53).

Our theory demonstrates that the idea of entanglement pointer, introduced in ref. 53 for fermions and bosons, can be non-trivially extended to the anyonic case, capturing the effect of braiding of Abelian quasiparticles, which manifests fractional statistics. Prospectively, our work unveils the relevance of statistics-induced entanglement to even more sophisticated settings. In particular, our work motivates further studies of Andreev-like tunneling beyond Laughlin quasiparticles, employing either particle-like or hole-like FQH fractions, as well as more exotic quasiparticles (like, e.g., "neutralons" and non-Abelian anyons), under non-equilibrium conditions. Although Andreev-like reflection has been investigated in various setups that comprise non-Abelian edge states[60–62], the highly intriguing challenge from this perspective would be to realize strongly diluted beams, facilitating the braiding of non-Abelian anyons with Andreev-reflected fractional quasiparticles. Given the present analysis, we expect that such braiding will generate entanglement induced by non-Abelian statistics. It is feasible (see Section IX of the SI) to extend our framework, which quantifies entanglement induced by Laughlin quasiparticle statistics, to non-Abelian systems. Generating and quantifying the statistics-induced entanglement through transport experiments will allow both the identification of non-Abelian states (cf. refs. 63,64 and references therein) and the manipulation of entanglement in topological quantum platforms. In particular, this may shed more light on the complex structure of non-Abelian edges—distinguishing between the candidate states—through the entanglement content obtained from transport noise measurements.

Finally, the statistics-induced entanglement reported here is evidently a topological phenomenon, since the anyonic fractional statistics is a manifestation of topology. Crucially, our Andreev entanglement pointer, given by Eqs. (4) and (3), explicitly vanishes at $\nu = 1$, which is a feature shared by the topological entanglement entropy[65]. It stands to reason to envision that employing our framework can provide direct access to topological entanglement entropy, which is expected to open up further avenues in experimentally studying systems with topological order.

## Methods
### Theoretical model
We consider the anyonic setup shown in Fig. 1, which consists of two source arms ($sA$, $sB$) and two middle ones ($A$, $B$). The system is viewed as comprising two subsystems, $\mathcal{A}$ (including $sA$ and $A$) and $\mathcal{B}$ ($sB$ and $B$). The system Hamiltonian contains the three parts: $H = H_{\text{arms}} + H_{\text{diluter}} + H_{\text{T}}$. The arms, carrying quasiparticles of charge $\nu e$, can be described by the bosonized edge Hamiltonian $H_{\text{arms}} = \nu \sum_\alpha \int dx [\partial_x \phi_\alpha(x)]^2 / 4\pi$, with $\phi_\alpha$ the bosonic field labeled by $\alpha = sA$, $sB$, $A$, $B$, see Fig. 1d. The dynamical bosonic phase obeys the standard commutation relation $[\partial_x \phi_\alpha(x), \phi_\beta(x')] = i\pi \delta_{\alpha\beta} \delta(x - x')$.

Fractional charges tunnel from sources to middle arms through the FQH bulk at two QPCs. These two diluter QPCs are described by the Hamiltonians written in terms of the anyon field operators $\psi_\alpha$: $H_{\text{diluter}} = \zeta_A \psi_A^\dagger \psi_{sA} + \zeta_B \psi_B^\dagger \psi_{sB} + \text{H.c.}$ Via bosonization, tunneling operators can be written through $\psi_\alpha = \exp(i\sqrt{\nu}\phi_\alpha)/(2\pi a)$, with $a$ an

ultraviolet cutoff (the smallest length scale in the problem). Strictly speaking, tunneling operators contain Klein factors that guarantee proper commutation relations for distant tunneling operators, which is important in systems that support circulating currents (e.g., in a Mach-Zehnder interferometer, see, e.g., refs. 33,66). The Klein factors are, however, irrelevant to the HOM setup, where currents cannot travel back and forth (cf. ref. 67), so that we do not introduce them here. The tunneling amplitudes $\zeta_A$ and $\zeta_B$ define the "bare" tunneling probabilities at the diluters, $\mathcal{T}_A^{(0)} = |\zeta_A|^2$ and $\mathcal{T}_B^{(0)} = |\zeta_B|^2$. The experimentally accessible transmission probabilities of diluters $\mathcal{T}_A$ and $\mathcal{T}_B$ are proportional to the corresponding bare probabilities: $\mathcal{T}_A \propto \mathcal{T}_A^{(0)}$ and $\mathcal{T}_B \propto \mathcal{T}_B^{(0)}$. We assume strong dilution, $\mathcal{T}_A^{(0)}, \mathcal{T}_B^{(0)} \ll 1$. In this work, the same voltage bias $V$ is assumed in both sources, and the single-source scenario is realized by pinching off either diluter.

The middle arms $A$ and $B$ communicate at the central QPC characterized by the bare transmission probability $\mathcal{T}_C^{(0)}$, which is related to the experimentally accessible transmission probabilities following $\mathcal{T}_C^{(0)} \propto \mathcal{T}_C / \sqrt{\mathcal{T}_A \mathcal{T}_B}$. The central QPC is placed at a distance $L$ from two diluters, in the downstream transport direction [Fig. 1a]. At variance with the two diluters, where the two depletion gates (the black area in Fig. 1b) are well separated in space, the central QPC is in the opposite limit, where the gates are almost "touching" each other (Fig. 1c). Following self-duality of tunneling through FQH QPCs (see, e.g., refs. 68–70), only fermionic tunneling is allowed in this limit. Physically, there is no bulk states with filling factor $\nu$ between the two arms (red and blue in Fig. 1), and, hence, between the subsystems $\mathcal{A}$ and $\mathcal{B}$, inside the central QPC. The tunneling at the central QPC is therefore described by the Hamiltonian $H_{\text{T}} = \zeta_C \Psi_A^\dagger \Psi_B + \text{H.c.}$, with $\zeta_C \propto \left[\mathcal{T}_C^{(0)}\right]^{1/2}$ and $\Psi_\alpha = \exp(i\phi_\alpha/\sqrt{\nu})/\sqrt{2\pi a}$ is the fermionic field operator. This bosonized expression contains $\sqrt{1/\nu}$ instead of $\sqrt{\nu}$ encountered above, which is a hallmark of electron tunneling in anyonic systems.

The building blocks of the entanglement pointer are current correlators. In the Andreev-tunneling limit at the collider, $\mathcal{T}_C^{(0)} \ll 1$, the noise of the current operator $\hat{I}_{\text{T}} = i\zeta_C \Psi_B^\dagger \Psi_A + \text{H.c.}$ is given by

$$S_{\text{T}} = e^2 \mathcal{T}_C^{(0)} \int dt \left\langle \left\{ \Psi_B^\dagger(0)\Psi_A(0), \Psi_A^\dagger(t)\Psi_B(t) \right\} \right\rangle_{\mathcal{T}_C^{(0)}=0}, \quad (8)$$

with $\{ , \}$ denoting an anticommutator. Evaluation of $S_{\text{T}}$, which yields Eqs. (2) and (3), involves correlators $\langle \Psi_A^\dagger(t)\Psi_A(0) \rangle$ and $\langle \Psi_B^\dagger(t)\Psi_B(0) \rangle$ at the position of the central QPC (see Section I of the SI). These correlation functions are greatly influenced by statistics of the quasiparticles involved, thus generating dependence on statistics in the observables—tunneling current and noise.

The evaluation of Eq. (7) requires explicit expression for the tunneling current,

$$I_{\text{T}} = e\mathcal{T}_C^{(0)} \int dt \left\langle \left[ \Psi_B^\dagger(0)\Psi_A(0), \Psi_A^\dagger(t)\Psi_B(t) \right] \right\rangle_{\mathcal{T}_C^{(0)}=0}, \quad (9)$$

with $[ , ]$ denoting a commutator. It is obtained using the correlation functions similar to those in Eq. (8) (see Section IIIB of the SI), yielding $I_{\text{T}} = I_{\text{T}}^{\text{single}} + I_{\text{T}}^{\text{collision}}$, where

$$I_{\text{T}}^{\text{single}} = \text{Re}\left\{ \mathcal{T}_C \mathcal{T}_A \frac{\nu e^2 V}{2} \frac{(2\pi\nu)^{1-\nu_s} e^{i\pi(\nu_s - \nu\nu_s + 1)}}{\pi\nu \sin(\pi\nu_s) + 2f_1(\nu)\mathcal{T}_A} [2i\pi\nu - \mathcal{T}_A(1 - e^{-2i\pi\nu})]^{\nu_s - 1} \right\} - \{A \to B\},$$

$$(10)$$

$$I_{\text{T}}^{\text{collision}} = \text{Im}\left\{ \mathcal{T}_C e^2 V \frac{\sqrt{\mathcal{T}_A \mathcal{T}_B} f_2(\nu) \sin(\pi\nu_d/2)}{\pi\nu \sin(\pi\nu_s) + 2f_1(\nu)\sqrt{\mathcal{T}_A \mathcal{T}_B}} [\mathcal{T}_A(1 - e^{-2i\pi\nu}) + \mathcal{T}_B(1 - e^{2i\pi\nu})]^{\nu_d - 1} \right\},$$

$$(11)$$

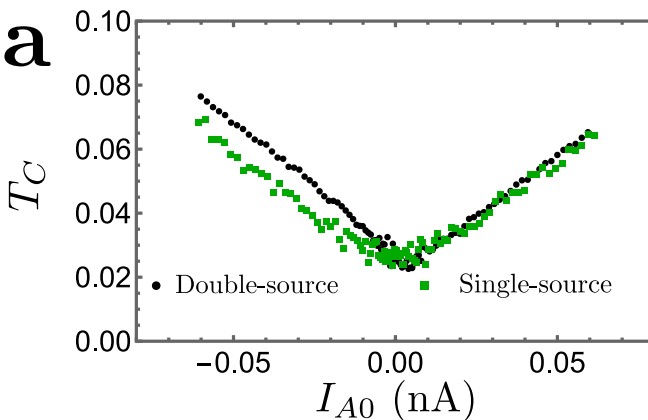

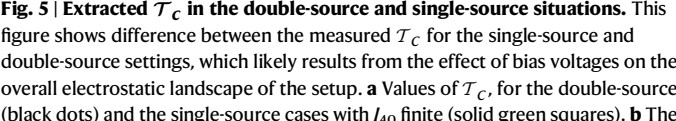

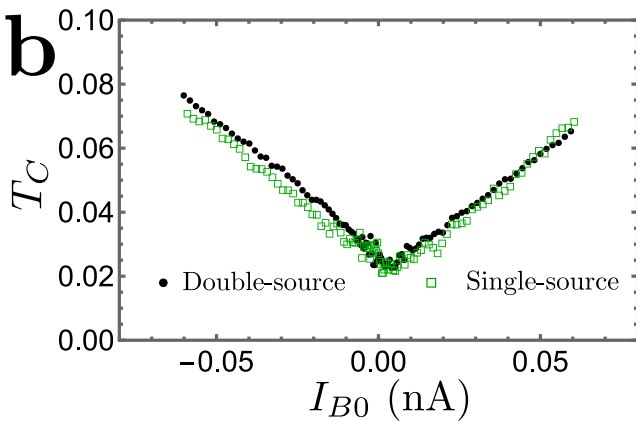

**Fig. 5 | Extracted $\mathcal{T}_C$ in the double-source and single-source situations.** This figure shows difference between the measured $\mathcal{T}_C$ for the single-source and double-source settings, which likely results from the effect of bias voltages on the overall electrostatic landscape of the setup. **a** Values of $\mathcal{T}_C$, for the double-source (black dots) and the single-source cases with $I_{A0}$ finite (solid green squares). **b** The same double-source transmission $\mathcal{T}_C$ (black dots) is compared to that of the single-source case with $I_{B0}$ finite (empty green squares). Corresponding values of $\mathcal{T}_A$ and $\mathcal{T}_B$ (arranging approximately between 0.02 and 0.08) are provided by the Supplementary Fig. S7.

with functions $f_{1,2}(v)$ defined in Eqs. (2) and (3). Compared to those expressions for the contributions to the tunneling noise $S_T$, the main difference of the tunneling currents from the corresponding noises is (apart from a trivial overall prefactor $1/e$) in their parity: "$-\{A \rightarrow B\}$" replaces "$+\{A \rightarrow B\}$" in the single-source term (10) and "Im" replaces "Re" in the collision term (11).

Note added: While preparing the first version of our manuscript, we noticed ref. 50, which concerned a single-source platform and did not address the effects of collisions and anyon-quasihole braiding.

## Experiment

The measurements are realized at $T \approx 35$ mK on a 2DEG set to $v = 1/3$. The device includes two nominally identical source QPCs positioned symmetrically with respect to a central QPC (see the SI and ref. 27). Gate voltages allow us to tune the QPCs in the configuration where the Andreev tunneling of quasiparticles takes place. The source QPCs are set in the anyonic-tunneling regime (Fig. 1b) and exhibit a shot-noise Fano factor corresponding to a fractional charge $e^* \approx e/3$, whereas the central QPC is tuned in the Andreev-like tunneling regime (Fig. 1c) with the tunneling charge $e^* \approx e$, as deduced from shot noise[27]. An experimental challenge is to be able to obtain reliably the entanglement pointer. Indeed, $\mathcal{P}_{\text{Andreev}}$ is a small difference between larger quantities measured separately, which increases the sensitivity to experimental artifacts such as drifts in time between compared configurations or unwanted small capacitive cross-talks. The difficulty is further enhanced by the difference between the measured $\mathcal{T}_C$ for the single-source and double-source settings (see fig. 5), which results, apparently, from the different electrostatic landscapes. As further detailed in Section VIII of the SI, the data set used to extract the entanglement pointer was obtained following a specific protocol reducing such artifacts. In particular, there are no changes in the device gates voltages, and the time between compared configurations is minimized. Further details on the experiment can be found in Section VII of the SI.

## Data availability

Raw data of this work can be accessed via Zenodo: https://doi.org/10.5281/zenodo.10434474.

## Code availability

Relevant Mathematica notebook can be accessed via Zenodo: https://doi.org/10.5281/zenodo.10434474.

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

## Acknowledgements

We are grateful to Gabriele Campagnano, Domenico Giuliano, Moty Heiblum, Thierry Martin, Bernd Rosenow, Inès Safi, and Kyrylo Snizhko for fruitful discussions. We thank O. Maillet, C. Piquard, A. Aassime, and A. Anthore for their contribution to the experiment. IG and YG acknowledge the support from the DFG grant No. MI658/10-2 and German-Israeli Foundation (GIF) grant No. I-1505-303.10/2019. YG acknowledges support from the Helmholtz International Fellow Award, by the DFG Grant RO 2247/11-1, by CRC 183 (project C01), the US-Israel Binational Science Foundation, and by the Minerva Foundation. GZ acknowledges the support from the startup grant at Nanjing University, National Natural Science Foundation of China (Grant No. 12374158) and Innovation Program for Quantum Science and Technology (Grant No. 2021ZD0302400). FP acknowledges the support of the European Research Council (ERC-2020-SyG-951451) and of the French RENATECH network.

## Author contributions

GZ, IG, and YG carried out the theoretical analysis. PG obtained the experimental data and performed the low-level data analysis under the supervision of FP. GZ designed and performed the data-theory comparison, with critical inputs from FP. All authors participated in the scientific discussions, contributed to the preparation of this work and to the writing of the paper, and proofread the manuscript.

## Funding

## Competing interests

The authors declare no competing interests.
