## [Transparent Peer Review file · Nature Communications]

Fractional-statistics-induced entanglement from Andreev-like tunneling

Corresponding Author: Professor Igor Gornyi

Version 0:

Reviewer comments:

Reviewer #1

(Remarks to the Author)

In this manuscript, the authors focus on the anyonic statistics in the context of recent experiments with quantum point contact (QPC) in the fractional quantum Hall (FQH) regime. In particular, the considered setup involves two side QPCs tuned to the weak-backscattering regime, which plays the role of anyon sources, and a central one where anyons impinge. This is becoming a standard setting in this context, but the authors focus on a specific case where the central QPC is tuned to the strong-backscattering, i.e. only electrons are allowed to tunnel since no FQH liquid connects the two sides of the central QPC. In this case, the physics at the QPC can be described in terms of some Andreev-like reflection of fractional quasiparticles, reminiscent of Andreev reflections involving splitting of Cooper pairs between a metal and a superconductor.

The main result presented by the authors is the definition of an entanglement pointer which is able to reveal the fractional statistics of anyons, even if only electron tunneling is allowed at the central QPC.

In my opinion, the interpretation provided by the authors is correct. They carefully analyze different elements to build up their interpretation and that is an aspect of their methodology that I appreciated a lot. Moreover, the theory is supported by experimental data which are properly analyzed without any unjustified claim.

Concerning the significance of this manuscript, I must acknowledge that the presence of experimental evidence is turning what could be just an interesting theoretical proposal into something that could impact the specific field of anyon physics in condensed matter. Having discovered this additional two-particle term might open new ways of interpreting previous and future experimental results. Indeed, since this contribution is estimated to be very small (sixth-order in perturbation theory), its experimental verification is crucial to assess its validity and significance. Anyway, I must admit that the significance of their results would probably not impact a general community since nothing significantly new about anyonic physics have been discovered.

The employed theoretical methodology is the standard one, combining the chiral Luttinger liquid theory to the Keldysh formalism, despite I have some comments on the calculations carried on in the SI, which I will specify later.

I cannot comment about experimental techniques, but the data analysis sounds valid and correct, also using standard definitions of quantities, such as transmission and current-current correlates.

Concerning the approach employed for the calculations, I have some technical questions, that I would like to see addressed by the authors.

1) while computing the electron correlation function in the first section of the SI, the authors make the following comment: « We assume that two out of four non-equilibrium vertex contract with electron operators at the central QPC ». I believe this assumption is quite strong and it should be further justified. Indeed, this correlator is a core element in the derivation of their results;

2) In Eq. (S43) the correction to the tunneling current noise due to the Coulomb interaction is presented. It is not clear to me where this term is supposed to enter into the definition of the entanglement pointer. Some paragraphs below Eq. (S43), the authors claim that this term is « removed » in the entanglement pointer, but to me it is not clear where this quantity enter in the entanglement pointer and which contribution is cancelling it. Could the author provide more information about it. ?

The main weak point of this manuscript is the clarity and the readability for a broad audience. The main text is far too technical and some parts use too much « jargon », which can be understood only by people from this community (i.e., quantum transport of anyons). I will make some examples where I believe the authors can improve the readability of the manuscript.

1) I would explicitly explain why in the strong-backscattering regime only electron tunneling is present;

2) Below Eq. (1), the authors write twice that the entanglement pointer can be written in terms of noise, first by pointing out to Eq. (6) and then to Eq.(5) and following discussions. Similarly, the two sentences « The entanglement pointer effectively subtracts out redundant contributions present when only one of the two sources is biased » and « Importantly, following its definition P_{Andreev} excludes the single-source contributions and contains only noise from the two-particle collisions. » seem to include the same information in my opinion. I would suggest the author to re-write and simplify the discussion around Fig.1 to improve its readability;

3) Despite not many calculations and equations are present, there are far too many technical terms in the main text. As an example, the authors mention « ultraviolet contributions », « time integrals » and « self-contracted pairs »: it is impossible to follow this discussion without constantly opening the SI to understand which « time integrals » they are talking about. At this stage I would suggest to employ a more intuitive and physical picture to describe what they have in mind. In this form, some paragraphs of the main text look like a short resume of the SI's sections and this negatively influences the clarity of the message.

In conclusion, despite the results obtained by the authors are relevant for the community interested in experiments involving anyons, I am not convinced that the present form of the manuscript can be interesting for a broader audience. I believe that a more pedagogical presentation of their results might also help to make the paper more appealing for a general audience. Unless the authors might consider to publish in more specialized journal, an extensive revision of the presentation of their results should be done before I can recommend the paper for publication in this journal.

Reviewer #2

(Remarks to the Author)

Entanglement is a fundamental property of a quantum system that hosts nonlocality of information. In the 20th century, the entanglement of photons was studied in quantum optics to establish the experimental foundations of quantum mechanics. Recent significant progress in measurement and control techniques of semiconductor quantum systems makes it possible to generate and evaluate entanglement in solid-state devices. Some of the authors of this paper have previously published methods for evaluating boson or fermion entanglement in a two-input collider setup. This manuscript discusses applying the statistics-induced entanglement pointer method in the previous study to a fractional quantum Hall setup with Andreev-like tunneling. The direction of applying the methods established in bosons and fermions to anyons is straightforward and an appropriate thematic setting based on natural scientific interest.

The statistics-induced entanglement pointer allows for the evaluation of entanglement due to the statistical nature of anyons by removing the effect of Coulomb interactions on the diluted anyon flow. The most interesting aspect of this manuscript is the application of this technique to a system with Andreev-like tunneling processes, i.e., systems coupled only by tunneling of fermions. In this setup, the two subsystems hosting anyons can be viewed as separated by a non-topological barrier. Nevertheless, this manuscript concludes that the topological quantum states of these systems, determined by anyon braiding, can be entangled. Although this result seems strange at first glance, it is possible due to the Andreev-like tunneling process peculiar to fractionally charged quasiparticles and is an interesting discovery. The authors compare the results of their theoretical calculations with experimental data and find good agreement between them, providing some support for the validity of their method.

I find the content of this manuscript very interesting and of a high scientific level. If the authors respond appropriately to the following comments, I recommend this paper for publication in Nature Communications.

1. I think the keyword anyonic entanglement is misleading. When I saw the word in the abstract, I was confused, as if the authors were describing an entanglement between anyons residing in two subsystems separated by a vacuum. Because it is the topological quantum state of the two subsystems that are entangled and not the anyons themselves, it may be better not to use this expression.

2. The figure captions or text should more carefully explain how the topological quantum state of the subsystems are determined by the array of anyons, and more plainly explain how entanglement is caused by Andreev-like or anyon tunnels. New figures could be added if they would aid the reader in understanding more quickly.

3. To allow the reader to correctly interpret the experimental data from the main text, key parameters, such as the transmission probability T_c of the central QPC, should be explicitly mentioned in the text and figure captions.

4. Section 5 compares the theoretical calculations with the experimental data and states that they are in good agreement. At the end of this section, there should be a sentence stating what a reader should take away from this agreement. If possible, it would be desirable to explain what the experimental and theoretical errors in Figure 3(b) mean.

Reviewer #3

(Remarks to the Author)

The authors have applied the entanglement pointer, which has been developed by part of the authors, to an anyonic system, successfully quantified the entanglement generated by quantum statistics, and showed that the theoretical prediction agrees well with the Hong-Ou-Mandel interferometer experiment. Though entanglement is a well-established concept in static few-body systems, that in dynamical many-body system is elusive and is a subtle problem. Anyonic systems are the platforms of quantum information processing and entanglement would have a pivotal role there. In light of recent flourishing of anyon research in fractional quantum Hall systems, I believe that extracting the entanglement content in an anyonic system both theoretically and experimentally would be significant in the field of both condensed matter physics and quantum information. The method looks appropriate as long as I understand and the result is reasonable.

On the other hand, it is difficult to say that the manuscript is easy to read for a broad readership. The focus of the present work is the Andreev-like tunneling, which, as the authors mentioned, is halfway between purely fermionic and anyonic colliders. Compared with the latter two simple systems, the setup including the Andreev-like tunneling and the mechanism behind it appears to be complicated for the purpose of quantifying the entanglement of anyons, and indeed it was difficult to understand how it works by the manuscript. In addition, perhaps because of the large number of relevant previous studies, comparisons with them are described in detail (e.g., connected and disconnected diagrams, ultraviolet and infrared contributions, equilibrium and non-equilibrium anyons) while the details of the calculation and what they mean in the present setup are left for the supplemental information. Though it is necessary to clarify the difference, they might have made not a small portion of the manuscript understandable only to those who are familiar with this field.

I am not sure if the above is the reason or not, but in my opinion it was difficult to find clear noteworthy findings from the manuscript beyond just applying the entanglement pointer to an anyonic system. It looks like a natural generalization of the authors' previous work and appears to be anticipated to some extent. Although the successful application of a new concept of the entanglement pointer to an anyonic system and the good agreement between the theory and experiment are major developments, I believe that the standard of the journal would require more impact than that. Specifically, it is somewhat questionable whether the present work will have a similar amount of innovation as the development of the entanglement pointer.

From general perspective, it is unclear how the entanglement studied here and its resilience to interactions are useful in quantum information processing. I think the entanglement studied here is generated by quantum statistics but is not topological since two anyons have to come closer and be scattered via electron tunneling. In this situation, entanglement from interaction might always accompany. So, I am confused whether it is legitimate to discriminate that by quantum statistics and by interaction for the purpose of quantum information processing.

In conclusion, I am reserved to recommend the manuscript for publication according to the acceptance criteria. I would rather like to recommend resubmission to a more specialized journal as the current presentation is not intended for a broad readership. Otherwise, I would like the authors to revise the manuscript so that it can be understood by a wider audience and then clearly state the impact of this work.

Version 1:

Reviewer comments:

Reviewer #1

(Remarks to the Author)

The authors provided satisfactory responses to my questions, and I greatly appreciated the politeness and clarity of their reply letter. The presentation of the manuscript has now reached the required standard and appropriately reflects the high quality of the results presented.

My only remaining concern pertains to the calculation of the correlator in Section IC of the Supplementary Information. While the explanation has been significantly improved, I still have some reservations regarding the approximations made with respect to the variable "t", which is integrated over the full range from $-\infty$ to $+\infty$ in the computation of transport quantities. In such a context, it is not rigorous to take limits over this variable. Nevertheless, the approximations employed by the authors are consistent with those found in other published works, and I acknowledge that they have presented them with commendable clarity, allowing the reader to follow the derivation in a transparent manner.

Taking all this into account, I recommend the manuscript for publication in Nature Communications without further modifications.

Reviewer #2

(Remarks to the Author)

I have reviewed their revised manuscripts and respect their thorough rewriting. The authors responded appropriately to my questions and comments. It was pointed out by the other reviewers that the manuscript was too complicated for general readers and, therefore, unlikely to attract their interest. I believe this issue has also been addressed. For these reasons, I

recommend that this manuscript be published in Nature Communications, as stated in my first review.

Reviewer #3

(Remarks to the Author)

Having read the revised manuscript and response, I believe that the readability of the manuscript has largely improved with clear conclusions, and all my comments and questions have been addressed adequately. The logic is easier to follow and the rich physics underlying their theory is easier to understand. I believe that the method invented by the authors for extracting the entanglement by the transport measurement would be invaluable for both fundamental condensed matter physics and quantum information. I recommend the manuscript for publication in Nature Communications without further revision.

RESPONSE TO REVIEWER 1

Reviewer 1: *In this manuscript, the authors focus on the anyonic statistics in the context of recent experiments with quantum point contact (QPC) in the fractional quantum Hall (FQH) regime. In particular, the considered setup involves two side QPCs tuned to the weak-backscattering regime, which plays the role of anyon sources, and a central one where anyons impinge. This is becoming a standard setting in this context, but the authors focus on a specific case where the central QPC is tuned to the strong-backscattering, i.e. only electrons are allowed to tunnel since no FQH liquid connects the two sides of the central QPC. In this case, the physics at the QPC can be described in terms of some Andreev-like reflection of fractional quasiparticles, reminiscent of Andreev reflections involving splitting of Cooper pairs between a metal and a superconductor.*

The main result presented by the authors is the definition of an entanglement pointer which is able to reveal the fractional statistics of anyons, even if only electron tunneling is allowed at the central QPC.

In my opinion, the interpretation provided by the authors is correct. They carefully analyze different elements to build up their interpretation and that is an aspect of their methodology that I appreciated a lot. Moreover, the theory is supported by experimental data which are properly analyzed without any unjustified claim.

Concerning the significance of this manuscript, I must acknowledge that the presence of experimental evidence is turning what could be just an interesting theoretical proposal into something that could impact the specific field of anyon physics in condensed matter. Having discovered this additional two-particle term might open new ways of interpreting previous and future experimental results. Indeed, since this contribution is estimated to be very small (sixth-order in perturbation theory), its experimental verification is crucial to assess its validity and significance.

We thank Reviewer 1 for carefully reviewing our work and presenting a concise summary. We are pleased to learn the overall positive assessment of our work by Reviewer 1, including the theoretical analysis and experimental evidence of the importance of anyonic collisions. In particular, we very much appreciate the comment regarding the significance of our work for interpreting previous and future experimental results in the vibrant field of anyon physics.

Reviewer 1 puts forward several concerns, questions, and suggestions. Below we respond to each of them (marked as A1-A7), one after another.

A1. *Anyway, I must admit that the significance of their results would probably not impact a general community since nothing significantly new about anyonic physics have been discovered.*

The employed theoretical methodology is the standard one, combining the chiral Luttinger liquid theory to the Keldysh formalism, despite I have some comments on the calculations carried on in the SI, which I will specify later. I cannot comment about experimental techniques, but the data analysis sounds valid and correct, also using standard definitions of quantities, such as transmission and current-current correlates.

Let us begin by highlighting two fundamental issues concerning our study. Appreciating these points will put the

importance and novelty of our results in the right perspective. We also note that this crucial comment by Reviewer 1 is closely related to the critique expressed by Reviewer 3 (see point C4 below).

1. Novel braiding mechanism – anyon-quasihole braiding

All the pioneering experiments so far that employed “anyonic collider platforms” and reported on signatures of fractional statistics, relied on so-called “vacuum bubble braiding” [Ref. 38: C. Han *et al.*, Nature Communications **7**, 11131 (2016)], a.k.a. “time domain braiding” [Ref. 36: B. Rosenow *et al.*, PRL **116**, 156802 (2016), Ref. 42: N. Schiller *et al.*, PRL **131**, 186601 (2023), and Ref. 46: J.-Y. M. Lee *et al.*, Nature Communications **13**, 6660 (2022)]. This phenomenon originates in the following process: an incoming anyon travels from the diluter and braids with a spontaneously generated anyon/anti-anyon pair, which then recombines and disappears (this is now schematically depicted in Figs. S3d to S3f of the latest version of our work).

Our work discovered a novel type of braiding phenomenon, i.e., anyon-quasihole braiding, which is induced by Andreev-like tunneling at the central QPC. Braiding between the quasihole, reflected after an Andreev-like tunneling, and anyons from the dilute beams, which bypass the central collider, is especially nontrivial, considering the fermionic nature of tunneling particles through the central collider. Further, anyon-quasihole braiding requires the presence of two “non-equilibrium” anyons: one triggering Andreev tunneling and the other braiding with the resulting quasihole. This is in great contrast to time-domain braiding for anyonic tunneling, where already a single anyon supplied from the diluter is sufficient, as the other anyon is spontaneously generated (within an anyon-hole pair) at the collider.

Unfortunately, in the original version of the manuscript, we did not stress these crucial differences between the two braiding phenomena, and even referred to anyon-quasihole braiding as time-domain braiding. The Reviewer’s comment motivated us to reconsider the interpretation of braiding processes responsible for the dependence of the transport observables on the braiding phase for the Andreev-like tunneling. We believe that the discovery of a very nontrivial braiding phenomenon is a significant step in anyonic physics.

In the revised version of the manuscript, we clearly state the novelty of anyon-quasihole braiding. For the convenience of the readers, the anyon-quasihole braiding process is illustrated in Fig. 2 of the main text of the latest version of our work. The key differences between anyon-quasihole and time-domain braiding processes, in terms of both physical interpretations and explicit expressions for correlation functions, are provided and thoroughly discussed in Sec. II and Fig. S3 of the Supplementary Information (SI).

2. Anyonic collisions generate statistics-induced entanglement that can be probed in transport

Importantly, the measured noise reported in experiments on anyonic colliders does not directly deliver information about collisions (and, hence, collision-induced braiding) of anyons from two uncorrelated (and, evidently, unentangled) sources. While the time-domain braiding leads to a signature of the statistical phase in current-current correlations, since there is no real anyonic collision in this process, there is no direct entanglement associated with time-domain braiding. Here, we are dealing with real braiding between initially unentangled non-equilibrium anyonic beams. This braiding renders the beams entangled. Not only does our platform generate entanglement, but we also

provide a tool (our entanglement pointer) to unequivocally identify it.

Moreover, as Reviewer 1 appreciated in the summary of our results, the two-particle collision term in the noise discovered in our work “may open new ways of interpreting previous and future experimental results.” This is particularly important in view of the common belief that anyonic collisions are irrelevant in HOM *colliders*: our work, thus, justifies the name (collider) for this platform.

In our study, two independent dilute anyonic beams meet at the central collider. They can exchange only fermions between them. It is the “spooky action” of such fermions that renders the anyonic beams entangled, invoking the signature of fractional statistics. In short, this is the first time genuine inter-anyon entanglement, generated via two-quasiparticle collisions, is reported and analyzed. Not only do we generate this entanglement, but with our entanglement pointer, we are able to observe it.

In addition to this fundamental novelty, we stress that our work is the first to report *anyonic entanglement* probed with transport measurements. Furthermore, the statistics-induced entanglement reported here is a topological phenomenon, since the anyonic statistics can be considered as a manifestation of topology. Crucially, our Andreev entanglement pointer, given by Eqs. (4) and (3) of the main text, explicitly vanishes at $\nu = 1$. This is a feature shared by the topological entanglement entropy proposed by A. Kitaev and J. Preskill, PRL **96**, 110404 (2006). It stands to reason to envision that employing our framework will provide direct access to topological entanglement entropy, as we now mention in the outlook as a natural direction for future research.

Last but not least, the theory part of our study represents the first comprehensive study of noise in an anyonic HOM collider with diluted beams and Andreev-like tunneling at the central QPC. This analysis appeared as a by-product of our search for non-trivial phenomena related to the generation of entanglement between anyonic subsystems; nevertheless, we believe that, taken at its face value, the study of noise in such systems significantly advances our understanding of anyonic physics, even without any further connections to exciting questions about statistics-induced entanglement.

We agree with Reviewer 1 in that a general community is nowadays strongly interested in anyonic physics. Therefore, we are confident that the significance of novel anyonic physics as described above will impact the general community of physicists.

Corresponding changes:

- In the revised version of our work, we have introduced a significant number of modifications to the abstract, introduction, and conclusions, in order to better stress the novelty and significance of our work.
- Throughout the manuscript, we now emphasize that the newly discovered anyon-quasihole braiding is different from the “conventional” time-domain braiding. We have added a new figure, Fig. 2 of the main text, to illustrate the anyon-quasihole braiding processes in anyonic systems with the Andreev-like tunneling.
- We now compare anyon-quasihole braiding with time-domain braiding in a new dedicated section (Sec. II) and Fig. S3 of the SI.

A2. While computing the electron correlation function in the first section of the SI, the authors make the following comment: “We assume that two out of four non-equilibrium vertex contract with electron operators at the central QPC”. I believe this assumption is quite strong and it should be further justified. Indeed, this correlator is a core element in the derivation of their results.

We would like to thank Reviewer 1 for carefully reading our manuscript and for asking this very important question. The calculation of the correlator mentioned by Reviewer 1 indeed relies on a crucial approximation that was not clearly justified in the original version. Below, we explain the validity of this approximation in the limit considered by our current work.

As has been pointed out by Reviewer 1, there are indeed several possibilities to “contract” the vertex operators (i.e., to choose equal or close time arguments for them) in the next-to-leading-order contribution to the correlation function we calculate. More specifically, there are three major contributions at this order:

- (i) Four operators of “non-equilibrium anyons” (supplied by the diluters) form two pairs, and operators in each pair contract with each other;
- (ii) All four operators contract with fermionic operators [i.e., $\Psi_A^\dagger(L, t^-)$ and $\Psi_A(L, 0^+)$ of Eq. (S11)] that represent tunneling at the central QPC;
- (iii) The situation considered by our work, where two out of four anyon operators self-contract, and the other two contract with fermionic operators that represent tunneling through the central QPC.

Among these options, the process described by option (i) yields a vanishing contribution to the correlator, as a tunneling fermion does not braid with non-equilibrium anyons. Processes (ii) and (iii) both lead to finite contributions. Motivated by the Reviewer’s question, in Sec. IC of the revised SI, we have provided all the details of the derivation of contributions from both these processes. Importantly, in process (ii), the time arguments of all four anyonic operators (i.e., s_1 to s_4) are strongly limited, as all of them should be correlated with fermionic tunneling events that occur at time moments 0 and t . This is in great contrast to process (iii), where the time arguments of two self-contracted anyonic operators can be freely chosen between $-L/v$ and $t - L/v$. Consequently, process (ii) has a smaller phase space in comparison to that of process (iii).

This fact is now presented explicitly in Fig. S1 of the latest SI. Indeed, here the phase space for process (ii) is concentrated around a single point, thus being much smaller than that of process (iii), which is dominated by an entire linear segment in the same plane of two time variables. This fact is further supported by comparing the corresponding integrals, Eq. (S17) for process (ii) and Eq. (S23) for process (iii). As shown by Eq. (S26), the ratio between the contributions of processes (ii) and (iii) to the correlation function is proportional to \mathcal{T}_A , the transmission probability for tunneling through the diluter. This ratio is negligible in the weak-tunneling limit, where the contribution from process (ii) thus becomes negligible. As we focus on the weak-tunneling limit (actually, \mathcal{T}_A is around 0.02 to 0.08, cf.

Fig. S7 of the latest version), the contribution of process (ii) can be safely neglected in our analysis, as was stated in the original version of the manuscript.

Corresponding changes:

- We have extensively rewritten the text describing the derivation of the “core” correlator after Eq. (S11) of the SI, explaining all possible contributions to the outcome.
- In Secs. IB and IC of the SI, we have provided detailed derivations of all major contributions to the integral of Eq. (S11).
- We have added two new figures, Figs. S1 and S2, to illustrate details of the evaluation of the integral determining the correlator.

A3. In Eq. (S43) the correction to the tunneling current noise due to the Coulomb interaction is presented. It is not clear to me where this term is supposed to enter into the definition of the entanglement pointer. Some paragraphs below Eq. (S43), the authors claim that this term is “removed” in the entanglement pointer, but to me it is not clear where this quantity enter in the entanglement pointer and which contribution is cancelling it. Could the author provide more information about it?

We thank Reviewer 1 for raising this important question. We agree that the statements made in connection with the role of interactions in the original version of the manuscript were unclear. In response to the Reviewer’s question, we have significantly modified the corresponding discussion in the main text and extended Sec. V of the SI.

Specifically, as detailed in Sec. VB of the SI in the revised manuscript, there are two major types of interaction effects on the correlation functions, noise, and the entanglement pointer: (i) inter-edge interaction (interaction between the channels that belong to different edges of the setup, which are connected by the central collider) and (ii) intra-edge interaction between channels within the same edge, which could be relevant to complex edges with multiple channels (e.g., to some high-order anyonic edges that host multiple co-propagating edge states). The second type of interaction is not realized for simple Laughlin filling fractions, including $\nu = 1/3$ studied experimentally in our work.

In the setup we focus on in the manuscript, the interaction occurs between the edges coupled by the central QPC. Since the region where the two edges come close to each other has a rather small spatial extension, the effect of such inter-edge interaction is weak, leading to small corrections to both cross-correlation and the entanglement pointer (see SI Sec. VB). Moreover, as discussed after Eq. (S78) and Eq. (S79), the inter-edge-interaction correction to the correlation functions is proportional to a high power of the dimensionless interaction strength ($\propto \delta_{\text{edge}}^5$) and, thus, is insignificant, under typical experimental conditions, even for longer parallel segments of the edges. Thus, for the inter-edge interaction, there is no cancellation of the interaction corrections to noises in the entanglement pointer: all such corrections are, however, small and do not really influence our findings.

The situation will be, however, different when considering systems with complex edges that contain multiple edge channels. Indeed, following our discussions at the end of Sec. VC in the SI, interactions in such setups may lead to a significant correction to the cross-correlation due to the so-called charge fractionalization. This correction, which may even exceed the interaction-free noise, is the same in the single-source and double-source current correlators. Therefore, it is avoided by the subtraction of the single-source noises from the double-source noise when evaluating the entanglement pointer according to Eq. (6).

Corresponding changes:

- We have significantly modified Sec. VB of the revised version of the SI, where we now carefully discuss the influences of different types of interactions.
- We have modified the discussion of the interaction effects at the end of Sec. 3 of the main text and in Sec. 6 (Conclusions).

A4. The main weak point of this manuscript is the clarity and the readability for a broad audience. The main text is far too technical and some parts use too much “jargon”, which can be understood only by people from this community (i.e., quantum transport of anyons). I will make some examples where I believe the authors can improve the readability of the manuscript.

We are extremely grateful to Reviewer 1 for pointing out the problems with the accessibility of the original version of the manuscript to a broad audience and, in particular, for noticing that we used ‘jargon’ that is unclear to researchers outside of the “anyonic community.” In response to this critique, we have greatly improved the readability of our manuscript. Major modifications are listed below. Below, we carefully describe the changes made to the manuscript regarding all the specific unclear points listed by Reviewer 1.

Corresponding changes:

- We have significantly modified the main text and removed all the ‘jargon’ expressions.
- In the revised SI, we have explained unclear terms whenever possible, thus making reading our technical discussion smoother to non-experts.

A5. I would explicit explain why in the strong-backscattering regime only electron tunneling is present;

Laughlin quasiparticles are only allowed to exist in the spatial region of samples with the corresponding filling fractions. In the weak-tunneling regime, the QPC and the two edges connected by the QPC have the same filling fraction. In this regime, Laughlin quasiparticles are allowed to tunnel across the QPC. In the Andreev-like tunneling regime studied by our current work, the QPC is, however, depleted into a vacuum state for anyons, where Laughlin quasiparticles are not allowed to exist. Consequently, in this regime, only fermions are allowed to tunnel across the QPC.

Corresponding changes:

- In the revised version of our work, we have added several lines in Sec. 2 “Entanglement pointer for Andreev-like tunneling” to explain why only fermions are allowed to tunnel through the central QPC.

A6. Below Eq. (1), the authors write twice that the entanglement pointer can be written in terms of noise, first by pointing out to Eq. (6) and that to Eq. (5) and following discussions. Similarly, the two sentences “The entanglement pointer effectively subtracts out redundant contributions present when only one of the two sources is biased” and “Importantly, following its definition P_{Andreev} excludes the single-source contributions and contains only noise from the two-particle collisions” seem to include the same information in my opinion. I would suggest the author to re-write and simplify the discussion around Fig. 1 to improve its readability;

We thank Reviewer 1 for pointing out the redundancy in our writing, which has been eliminated in the new version. Following the Reviewer’s suggestion, we have simplified the corresponding discussion.

Corresponding changes:

- We have revised the discussion below Eq. (1), and, in particular, removed the reference to Eq. (6) after Eq. (1).

A7. Despite not many calculations and equations are present, there are far too many technical terms in the main text. As an example, the authors mention “ultraviolet contributions”, “time integrals” and “self-contracted pairs”: it is impossible to follow this discussion without constantly opening the SI to understand which “time integrals” they are talking about. At this stage I would suggest to employ a more intuitive and physical picture to describe what they have in mind. In this form, some paragraphs of the main text look like a short resume of the SI’s sections and this negative influences the clarity of the message.

We are grateful Reviewer 1 for the above suggestions and comments, which allowed us to substantially improve the transparency of our work to a broad audience. In the revised manuscript, we have made every effort to ensure that the terminology has been clearly explained.

Corresponding changes:

- Following the Reviewer’s recommendation, we have included more intuitive and physical pictures of the phenomena we analyze. In particular, we have added new figures to illustrate anyon-quasihole braiding and the processes responsible for inducing current-current correlations (leading to the generation of entanglement between initially unentangled anyonic beams), new panels of Fig. 2 and new Fig. 3 of the revised version. We have also extended the qualitative discussion in Sec. 4, which is now entitled “Physical interpretation of the entanglement pointer.” This makes the main text self-contained, such that consulting the SI is not necessary for comprehending the main ideas of the work.
- To increase the readability, we describe the newly introduced anyon-quasihole braiding process in an intuitive way in Sec. 2 of the main text. This process is also compared in detail to time-domain braiding in Sec. II of the latest Supplementary Information, where we included new Fig. S3 illustrating the two phenomena.

- In the previous version of our manuscript, we used the notions of “ultraviolet” and “infrared” to discuss different contributions to the integrals over time arguments of operators determining the expansions of the correlation functions (“time integrals” as they were referred to in the previous version of our manuscript). We agree with Reviewer 1 that this terminology involves technical aspects that are unfamiliar to non-experts, especially to experimentalists. We have removed these terms in the latest version of the main text and significantly simplified the corresponding discussion.
- In addition, the notions of “connected” and “disconnected” diagrams are removed altogether from the latest version of our work. In fact, based on our communications with colleagues, this jargon terminology was considered somewhat misleading even for experts from the “anyonic community,” so that avoiding it clearly enhanced the accessibility of our arguments.
- The same concerns with the notion of “contraction” and “self-contracted pairs” which was used extensively in the SI, where it could be confused with conventional contractions within Wick’s theorem (not applicable to anyons). We have consistently replaced “contraction” with “contract” (which is similar but not equivalent), and carefully explained this invented terminology in Sec. IB.
- We also agree with comment C3 by Reviewer 3 (see below) that “non-equilibrium anyon” is another jargon term that is unclear to the readers. In the latest version of our work (including Fig. 1d and the beginning of Sec. 3 “Tunneling current noise”), we briefly explain that non-equilibrium anyons in each channel (corresponding to channel A or B of Fig. 1) are produced via anyonic tunneling through the corresponding diluter.

In conclusion, despite the results obtained by the authors are relevant for the community interested in experiments involving anyons, I am not convinced that the present form of the manuscript can be interesting for a broader audience. I believe that a more pedagogical presentation of their results might also help to make the paper more appealing for a general audience. Unless the authors might consider to publish in more specialized journal, an extensive revision of the presentation of their results should be done before I can recommend the paper for publication in this journal.

We thank Reviewer 1 once again for the very useful comments and recommendations. As detailed in our responses above, we have extensively rewritten the manuscript and have made revisions that address all these points. This has allowed us to significantly improve the clarity of presentation, to underscore the novelty and significance of our work, and to enhance the broad interest of potential readers. In particular, we have added further pedagogical exposition and clarification on the importance of our findings, which makes the revised manuscript of immediate interest to the broad readership of Nature Communications. We hope that our responses above and the corresponding changes made to the manuscript will be convincing for Reviewer 1.

RESPONSE TO REVIEWER 2

Reviewer 2: *Entanglement is a fundamental property of a quantum system that hosts nonlocality of information. In the 20th century, the entanglement of photons was studied in quantum optics to establish the experimental foundations of quantum mechanics. Recent significant progress in measurement and control techniques of semiconductor quantum systems makes it possible to generate and evaluate entanglement in solid-state devices. Some of the authors of this paper have previously published methods for evaluating boson or fermion entanglement in a two-input collider setup. This manuscript discusses applying the statistics-induced entanglement pointer method in the previous study to a fractional quantum Hall setup with Andreev-like tunneling. The direction of applying the methods established in bosons and fermions to anyons is straightforward and an appropriate thematic setting based on natural scientific interest.*

The statistics-induced entanglement pointer allows for the evaluation of entanglement due to the statistical nature of anyons by removing the effect of Coulomb interactions on the diluted anyon flow. The most interesting aspect of this manuscript is the application of this technique to a system with Andreev-like tunneling processes, i.e., systems coupled only by tunneling of fermions. In this setup, the two subsystems hosting anyons can be viewed as separated by a non-topological barrier. Nevertheless, this manuscript concludes that the topological quantum states of these systems, determined by anyon braiding, can be entangled. Although this result seems strange at first glance, it is possible due to the Andreev-like tunneling process peculiar to fractionally charged quasiparticles and is an interesting discovery. The authors compare the results of their theoretical calculations with experimental data and find good agreement between them, providing some support for the validity of their method.

I find the content of this manuscript very interesting and of a high scientific level. If the authors respond appropriately to the following comments, I recommend this paper for publication in Nature Communications.

We thank Reviewer 2 for carefully reviewing our work and providing a concise summary of our findings. We are pleased to learn that Reviewer 2 finds our manuscript “very interesting and of high scientific level” and recommends the publication (conditioned on our response to the comments below).

Reviewer 2 raises several crucial questions concerning our work, which we denote by $B1$ to $B4$ in our response below.

$B1$. *I think the keyword anyonic entanglement is misleading. When I saw the word in the abstract, I was confused, as if the authors were describing an entanglement between anyons residing in two subsystems separated by a vacuum. Because it is the topological quantum state of the two subsystems that are entangled and not the anyons themselves, it may be better not to use this expression.*

We thank the Reviewer for pointing out the ambiguity of the term “anyonic entanglement” used in the abstract. We have changed this term to “entanglement between anyonic beams.” Further in the text, we also use the notion of

“entanglement between the two subsystems.”

Corresponding changes:

- Following the Reviewer’s suggestion, we have changed “anyonic entanglement” to the more precise and adequate terms throughout the manuscript.

B2. The figure captions or text should more carefully explain how the topological quantum state of the subsystems are determined by the array of anyons, and more plainly explain how entanglement is caused by Andreev-like or anyon tunnels. New figures could be added if they would aid the reader in understanding more quickly.

We thank Reviewer 2 for these suggestions that clearly enhance the transparency of our work. In the revised version, we have introduced the corresponding changes (see also our response to question A7 by Reviewer 1).

Corresponding changes:

- As suggested by Reviewer 2, figure captions have been extended to make the figures self-contained.
- After Eq. (1), we have added an extra paragraph to explain the nature of the array of anyons in channel A .
- In the same paragraph, we also explain the generation of entanglement by a single Andreev-like reflection process.
- To increase the readability, we describe the newly introduced anyon-quasihole braiding process in an intuitive way in Sec. 2 of the main text. This process is also compared in detail to time-domain braiding in Sec. II of the latest Supplementary Information, where we included new Fig. S3 illustrating the two phenomena.
- We have added extra figures to illustrate the non-equilibrium anyonic array (Fig. 1d of the latest version) and the generation of entanglement (Fig. 3 of the latest version).

B3. To allow the reader to correctly interpret the experimental data from the main text, key parameters, such as the transmission probability T_C of the central QPC, should be explicitly mentioned in the text and figure captions.

We thank Reviewer 2 for this important suggestion. Corresponding changes have been introduced to the caption of Fig. 4 of the latest version of the manuscript.

Corresponding changes:

- In the caption of Fig. 4 of the latest version, we have added the reference to Fig. 5 and Fig. S7 of the Supplementary Information, where the data of T_C and T_A, T_B are provided, respectively.
- In the caption of Fig. 5 of the latest main text, we have stated the range of T_A and T_B .

B4. Section 5 compares the theoretical calculations with the experimental data and states that they are in good agreement. At the end of this section, there should be a sentence stating what a reader should take away from this agreement. If possible, it would be desirable to explain what the experimental and theoretical errors in Figure 3(b) mean.

We thank Reviewer 2 for this important suggestion (we note that Fig. 3 of the previous version is Fig. 4 of the revised one).

The remarkable theory-experiment agreement conveys the following two major messages: (i) The overall picture behind the physics observed, especially concerning the role of anyon-quasihole braiding, is valid in an anyonic system in the Andreev-like tunneling regime; (ii) The role of inter-edge interactions is insignificant in setups of this type.

The small deviation of experimental points from the theoretical predictions comes from the influence of thermal fluctuation on the experimental data.

Corresponding changes:

- We have added several lines to discuss the message conveyed by the theory-experiment agreement, right before Sec. 6 “Conclusions”. A brief explanation of the small deviation is included in the caption of Fig. 4 of the latest version.

RESPONSE TO REVIEWER 3

Reviewer 3: *The authors have applied the entanglement pointer, which has been developed by part of the authors, to an anyonic system, successfully quantified the entanglement generated by quantum statistics, and showed that the theoretical prediction agrees well with the Hong-Ou-Mandel interferometer experiment. Though entanglement is a well-established concept in static few-body systems, that in dynamical many-body system is elusive and is a subtle problem. Anyonic systems are the platforms of quantum information processing and entanglement would have a pivotal role there. In light of recent flourishing of anyon research in fractional quantum Hall systems, I believe that extracting the entanglement content in an anyonic system both theoretically and experimentally would be significant in the field of both condensed matter physics and quantum information. The method looks appropriate as long as I understand and the result is reasonable.*

We thank Reviewer 3 for carefully reviewing our paper and summarizing the essence of our work. We also greatly appreciate positive comments by Reviewer 3, especially that “extracting the entanglement content in an anyonic system both theoretically and experimentally would be significant in the field of both condensed matter physics and quantum information” and “the result is reasonable.”

Reviewer 3 raises several points that we mark *C1* to *C5* below.

C1. On the other hand, it is difficult to say that the manuscript is easy to read for a broad readership

We have followed the Reviewer’s critical comments and introduced significant changes to the manuscript, which, we feel, make the presentation accessible to a broad audience and, at the same time, clearly state the new physics and how our results differ from those of recent studies of anyonic systems. In particular, to enhance the transparency of physical discussions, we have removed the specialized terms from the main text and Methods sections. Three extra figures (i.e., Figs. 1d, 2, and 3 of the latest version) are also added to explain, respectively, (i) generation of highly non-equilibrium anyonic beams, (ii) the physical meaning of the newly introduced concept of anyon-quasihole braiding (describing the corresponding sequence of tunneling processes), and (iii) how the current correlations and entanglement are generated by an Andreev-like tunneling. The difference between anyon-quasihole braiding of our work and time-domain braiding of previous works is further illustrated by Fig. S3 of the latest Supplementary Information (SI).

We note that this comment by Reviewer 3 has much in common with the comments of the two other Reviewers (see our responses to questions A7 and B2 above). We admit that the original version of our manuscript suffered from a too strong focus on technical aspects, which overshadowed the main conceptual messages (nicely summarized by Reviewer 3 in the summary of our work). We believe that the extensive revisions made to the manuscript in response to the comments by all the Reviewers significantly improved accessibility of our paper to a broad readership.

C2. The focus of the present work is the Andreev-like tunneling, which, as the authors mentioned, is halfway between purely fermionic and anyonic colliders. Compared with the latter two simple systems, the setup including the Andreev-like tunneling and the mechanism behind it appears to be complicated for the purpose of quantifying the entanglement

of anyons, and indeed it was difficult to understand how it works by the manuscript.

As far as the characterization of the Andreev-like platform as being halfway between purely fermionic and anyonic colliders, we would like to stress two important points: First, to date, all studies of anyonic colliders reported the outcome of “time-domain” braiding and not genuine braiding of “colliding” anyons. Our study relies on two anyonic beams impinging on the collider. Second, given the Andreev-like platform, only fermions can tunnel between the two beams (as is the case with fermionic colliders). Remarkably, such tunneling fermions render the participating anyons entangled (a “spooky impact” by the fermions). Importantly, since the tunneling of fermions does not produce the braiding phase, the time-domain braiding mechanism, which “screens out” other braiding processes in anyonic colliders, is not effective in the Andreev-tunneling regime.

Below, we would like to clarify that our setup, as well as the mechanism behind our findings, is not more complicated than that of the anyonic collider from both theory and experiment perspectives. To begin with, we emphasize that our experimental setup, investigating Andreev-like tunneling, is basically the same Hong-Ou-Mandel (HOM) interferometer in recent works on anyonic colliders, Refs. [16-19]. As the only difference, being depleted by a local gate, the central QPC of our setup allows only fermions to tunnel. We stress that depleting local potential at the central QPC does not enhance the experimental complexity, as it is a well-established technique. The Andreev-like tunneling was already realized experimentally in such a setup in Ref. [27].

On the theoretical side, our setup comprises the Hong-Ou-Mandel interferometer structure of arms with anyonic diluters and the central QPC that allows only fermions to tunnel. The first ingredient is the same as that of anyonic collider structures. The second ingredient is also not more complicated than that of anyonic colliders in terms of the theoretical modeling within the bosonization approach. Actually, for a two-edge setup (without diluters), it is known that the Andreev-like tunneling regime is the fixed point that is dual to the anyon-tunneling one (see, e.g., the book “Quantum Dissipative Systems” by Ulrich Weiss). Mathematically, transformations that link tunneling operators in these two regimes are also well known and were employed in a number of previous theoretical works quoted in our manuscript. In addition, for both Andreev-like and anyonic HOM setups (the latter is still under investigation by us), the entanglement pointer quantifies the entanglement generated by quasiparticle statistics, which is manifested by the outcome of two-particle collisions (as stated in response to question A1), in both systems. From these perspectives, the Andreev-like HOM setup is of the same complexity as that with anyonic tunnelings, both theoretically and experimentally.

Nevertheless, we would like to point out that our research indeed grasps features that are novel and unique to Andreev-like tunneling systems. Actually, the entanglement pointer obtained in the Andreev-like tunneling regime is positive, different from the fermionic case, Ref. [53], where the entanglement pointer was negative. This result is highly non-trivial, as both Andreev-like tunneling and fermionic setups allow only fermions to tunnel through the central QPC. The change in sign of the entanglement pointer of an Andreev-like tunneling regime arises from the anyon-quasihole braiding between an anyon supplied by the diluter and the fractional quasihole Andreev-reflected at

the central QPC. The braiding phase that accompanies fermionic tunneling is certainly non-trivial (being physically different from the time-domain braiding, see Sec. II and Fig. S3 of the latest SI) and has not yet been noticed in the literature (see also our response to comment A1 by Reviewer 1). Thus, suppressing time-domain braiding by resorting to Andreev-like tunneling opens the way to investigate alternative possible braiding mechanisms that are concealed by time-domain braiding in more conventional anyonic colliders. These competing mechanisms can be expected to be more pronounced in anyonic colliders (and other platforms) with a different structure of the edges.

In the revised manuscript, we have more clearly stated that the Andreev-like tunneling setup has the same complexity as that of anyon-tunneling setups, both experimentally and theoretically.

Corresponding changes:

- We have added Fig. 1d to explain the generation of highly non-equilibrium anyonic beams.
- We have added Fig. 2, to illustrate a single Andreev-like tunneling process, including the anyon-quasihole braiding induced by the reflected fractional-charge hole.
- We have added Fig. S3 to compare anyon-quasihole braiding, which is introduced in our present work, and time-domain braiding of previous literature.
- In Sec. 1 “Introduction”, we compare Andreev-like tunneling with direct anyonic tunneling to show that the two setups have the same experimental complexity.

C3. In addition, perhaps because of the large number of relevant previous studies, comparisons with them are described in detail (e.g., connected and disconnected diagrams, ultraviolet and infrared contributions, equilibrium and non-equilibrium anyons) while the details of the calculation and what they mean in the present setup are left for the supplemental information. Though it is necessary to clarify the difference, they might have made not a small portion of the manuscript understandable only to those who are familiar with this field.

We acknowledge and fully agree with this important comment made by Reviewer 3. In the latest version of our work, we have moved most of the comparisons to previous works to Sec. II of the SI, keeping only the intuitive physical picture illustrating the difference between the braiding phenomena, such that the revised main text appears much more friendly to readers outside this field. Note that this comment by Reviewer 3 parallels comment A7 by Reviewer 1, so that our response to A7 also applies here.

Corresponding changes:

- In the main text, we have removed most of the discussions concerning the explicit comparison between our work and previous related studies. Instead, these materials are now added to Sec. II of the latest SI.
- A new figure, Fig. S3 has been added to the SI to better illustrate the difference between anyon-quasihole braiding in Andreev-like tunneling systems and time-domain braiding of anyon-tunneling platforms.

C4. I am not sure if the above is the reason or not, but in my opinion it was difficult to find clear noteworthy findings from the manuscript beyond just applying the entanglement pointer to an anyonic system. It looks like a natural generalization of the authors' previous work and appears to be anticipated to some extent. Although the successful application of a new concept of the entanglement pointer to an anyonic system and the good agreement between the theory and experiment are major developments, I believe that the standard of the journal would require more impact than that. Specifically, it is somewhat questionable whether the present work will have a similar amount of innovation as the development of the entanglement pointer.

We thank Reviewer 3 for this important question, which demonstrates that the original version of our manuscript suffered from the absence of clear-cut arguments showing that our study is novel and impactful. Since this crucial comment by Reviewer 3 concerning the novelty and significance of our study is very similar to comment A1 by Reviewer 1, we would like to repeat here our arguments from the response to A1. For the Reviewer's and Editor's convenience, we copy our response here.

Let us begin by highlighting two fundamental issues concerning our study. Appreciating these points will put the importance and novelty of our results in the right perspective.

1. Novel braiding mechanism – anyon-quasihole braiding

All the pioneering experiments so far that employed “anyonic collider platforms” and reported on signatures of fractional statistics, relied on so-called “vacuum bubble braiding” [Ref. 38: C. Han *et al.*, Nature Communications **7**, 11131 (2016)], a.k.a. “time domain braiding” [Ref. 36: B. Rosenow *et al.*, PRL **116**, 156802 (2016), Ref. 42: N. Schiller *et al.*, PRL **131**, 186601 (2023), and Ref. 46: J.-Y. M. Lee *et al.*, Nature Communications **13**, 6660 (2022)]. This phenomenon originates in the following process: an incoming anyon travels from the diluter and braids with a spontaneously generated anyon/anti-anyon pair, which then recombines and disappears (this is now schematically depicted in Figs. S3d to S3f of the latest version of our work).

Our work discovered a novel type of braiding phenomenon, i.e., anyon-quasihole braiding, which is induced by Andreev-like tunneling at the central QPC. Braiding between the quasihole, reflected after an Andreev-like tunneling, and anyons from the dilute beams, which bypass the central collider, is especially nontrivial, considering the fermionic nature of tunneling particles through the central collider. Further, anyon-quasihole braiding requires the presence of two “non-equilibrium” anyons: one triggering Andreev tunneling and the other braiding with the resulting quasihole. This is in great contrast to time-domain braiding for anyonic tunneling, where already a single anyon supplied from the diluter is sufficient, as the other anyon is spontaneously generated (within an anyon-hole pair) at the collider.

Unfortunately, in the original version of the manuscript, we did not stress these crucial differences between the two braiding phenomena, and even referred to anyon-quasihole braiding as time-domain braiding. The Reviewer's comment motivated us to reconsider the interpretation of braiding processes responsible for the dependence of the transport observables on the braiding phase for the Andreev-like tunneling. We believe that the discovery of a very nontrivial braiding phenomenon is a significant step in anyonic physics.

In the revised version of the manuscript, we clearly state the novelty of anyon-quasihole braiding. For the convenience of the readers, the anyon-quasihole braiding process is illustrated in Fig. 2 of the main text of the latest version of our work. The key differences between anyon-quasihole and time-domain braiding processes, in terms of both physical interpretations and explicit expressions for correlation functions, are provided and thoroughly discussed in Sec. II and Fig. S3 of the Supplementary Information (SI).

2. Anyonic collisions generate statistics-induced entanglement that can be probed in transport

Importantly, the measured noise reported in experiments on anyonic colliders does not directly deliver information about collisions (and, hence, collision-induced braiding) of anyons from two uncorrelated (and, evidently, unentangled) sources. While the time-domain braiding leads to a signature of the statistical phase in current-current correlations, since there is no real anyonic collision in this process, there is no direct entanglement associated with time-domain braiding. Here, we are dealing with real braiding between initially unentangled non-equilibrium anyonic beams. This braiding renders the beams entangled. Not only does our platform generate entanglement, but we also provide a tool (our entanglement pointer) to unequivocally identify it.

Moreover, as Reviewer 1 appreciated in the summary of our results, the two-particle collision term in the noise discovered in our work “may open new ways of interpreting previous and future experimental results.” This is particularly important in view of the common belief that anyonic collisions are irrelevant in HOM *colliders*: our work, thus, justifies the name (collider) for this platform.

In our study, two independent dilute anyonic beams meet at the central collider. They can exchange only fermions between them. It is the “spooky action” of such fermions that renders the anyonic beams entangled, invoking the signature of fractional statistics. In short, this is the first time genuine inter-anyon entanglement, generated via two-quasiparticle collisions, is reported and analyzed. Not only do we generate this entanglement, but with our entanglement pointer, we are able to observe it.

In addition to this fundamental novelty, we stress that our work is the first to report *anyonic entanglement* probed with transport measurements. Furthermore, the statistics-induced entanglement reported here is a topological phenomenon, since the anyonic statistics can be considered as a manifestation of topology.

Last but not least, the theory part of our study represents the first comprehensive study of noise in an anyonic HOM collider with diluted beams and Andreev-like tunneling at the central QPC. This analysis appeared as a by-product of our search for non-trivial phenomena related to the generation of entanglement between anyonic subsystems; nevertheless, we believe that, taken at its face value, the study of noise in such systems significantly advances our understanding of anyonic physics, even without any further connections to exciting questions about statistics-induced entanglement.

Corresponding changes:

- In the revised version of our work, we have introduced a significant number of modifications to the abstract, introduction, and conclusions, in order to better stress the novelty and significance of our work.

- Throughout the manuscript, we now emphasize that the newly discovered anyon-quasihole braiding is different from the “conventional” time-domain braiding. We have added a new figure, Fig. 2 of the main text, to illustrate the anyon-quasihole braiding processes in anyonic systems with the Andreev-like tunneling.
- We now compare anyon-quasihole braiding with time-domain braiding in a new dedicated section (Sec. II) and Fig. S3 of the SI.

C5. From general perspective, it is unclear how the entanglement studied here and its resilience to interactions are useful in quantum information processing. I think the entanglement studied here is generated by quantum statistics but is not topological since two anyons have to come closer and be scattered via electron tunneling. In this situation, entanglement from interaction might always accompany. So, I am confused whether it is legitimate to discriminate that by quantum statistics and by interaction for the purpose of quantum information processing.

We thank Reviewer 3 for this highly important question. We would begin by pointing out that our entanglement pointer is indeed topological. Crucially, our Andreev entanglement pointer, given by Eqs. (4) and (3) of the main text, explicitly vanishes at $\nu = 1$. This is a feature shared by the topological entanglement entropy proposed by A. Kitaev and J. Preskill, PRL **96**, 110404 (2006). It stands to reason to envision that employing our framework will provide direct access to topological entanglement entropy, as we now mention in the outlook as a natural direction for future research.

More specifically, in our case, Andreev-like tunneling triggered by two-anyon collisions depends on two quasi-particle features: (i) bunching or anti-bunching of two colliding particles and (ii) anyon-quasihole braiding between non-equilibrium anyons and reflected fractional-charge holes generated at the central QPC. The former feature, i.e., two-particle collision outcome, is known [see, e.g., “Shot noise in mesoscopic conductors” by Y. M. Blanter and M. Buttiker for fermionic and bosonic collision, and G. Campagnano et al, Phys. Rev. Lett. **109**, 106802 (2012) for anyonic collision] to be predetermined by the statistics of quasiparticles. We would like to stress that the fractional statistics is a *topological feature* that is free from the effect of any local interactions. The fact that the colliding particles are to be found closer to each other at the collider is imposed by the geometry of our setup, but does not spoil the topological character of the induced correlations. The anyon-quasihole braiding is also determined by the topology, or filling fraction, of fractional-charge edges, as long as edge states are distant from each other (i.e., when they are free from inter-edge interaction).

As shown after Eqs. (S92), (S93) of the SI, and the response to question A3 from Reviewer 1, the topological protection of entanglement pointer can be manifested by the fact that the entanglement pointer is resilient against intra-edge interactions along each transport edge. From the physical perspective, intra-edge interaction along transport edges will not influence the outcome of two-particle collisions (which generates the entanglement pointer) that occurs at the central QPC. Mathematically, this resilience feature of entanglement pointer is especially manifest, when comparing intra-edge interaction-induced influence on entanglement pointer [i.e., $\delta\mathcal{P}_{\text{Andreev}}^{\text{coulomb}}$ of Eq. (S94)], to

that on correlation functions [i.e., $\delta S_T^{\text{coulomb}}$ of Eq. (S93)]. As shown in the response to A3, and analyzed below Eq. (S93) of the SI, the correction induced by intra-edge interactions in the correlation functions may even exceed the interaction-free correlation, given strong enough dilutions. Considering this fact, the entanglement pointer is more robust against intra-edge interactions than correlation functions alone. It is thus more reasonable and legitimate to choose the entanglement pointer to characterize the statistics-induced quantities (including entanglement content) in quantum information processing, rather than correlation functions alone.

In the revised manuscript, we have more clearly stated the topological nature of the entanglement pointer, as well as its topologically protected resilience against interaction within the edges.

Corresponding changes:

- After Eq. (4) of the main text, we have added several lines to stress that our entanglement pointer vanishes when $\nu = 1$. This function thus discloses entanglement that is generated by anyonic collision and the anyon-quasihole braiding processes.
- A discussion is added after Eq. (S46) of the latest SI.
- We have significantly modified Sec. VB of the revised version of the SI, where we now carefully discuss the influences of different types of interactions.
- We have modified the discussion of the interaction effects at the end of Sec. 3 of the main text and in Sec. 6 (Conclusions).
- We have significantly modified the introduction and conclusions to stress the significance of topological facets of statistics-induced entanglement.

In conclusion, I am reserved to recommend the manuscript for publication according to the acceptance criteria. I would rather like to recommend resubmission to a more specialized journal as the current presentation is not intended for a broad readership. Otherwise, I would like the authors to revise the manuscript so that it can be understood by a wider audience and then clearly state the impact of this work.

We thank Reviewer 3 for all the suggestions and questions that have definitely helped us improve the transparency in our manuscript. We have taken all the critiques by Reviewer 3 into account in the latest version of our work, which has been significantly revised based on the Reviewer's comments. As detailed in our responses above, we have added a pedagogical exposition and clarification on the importance of our findings, which makes the revised manuscript of immediate interest to the broad readership of Nature Communications. We hope that our responses above and the corresponding changes made to the manuscript will be convincing for Reviewer 3.

ADDITIONAL MODIFICATION

Finally, we would like to draw the attention of all three Reviewers to the fact that in Section 5 (“Comparison with experiment”), we have now introduced a modification (not inspired by any of the Reviewers’ reports). In short, as now detailed in Supplemental Information, we have carried out a deeper analytical study of specific properties of the correlation functions and integrals underlying observable quantities, going beyond the approximations used in the original version. This has significantly streamlined the evaluation of experimentally measurable quantities. With this improvement, the previous adjustments employed to fit theory and experiment are no longer needed—experimental data now agree with the theoretical predictions with only minimal, if any, rescaling of the data.

I. LIST OF CHANGES

Based on the Referee Reports, we have introduced major modifications below, to the latest version of our work.

- We have improved the *Abstract*, where we have introduced the concept “anyon-quasihole braiding”, indicating a novel braiding-generating process that is unique for systems in the Andreev-like tunneling limit.
- We have extensively rewritten the *Introduction*. The latest version more carefully summarizes innovative advances of our work. Here we especially emphasize on the importance of the generation of anyonic-statistics induced entanglement, though only fermions are allowed to tunnel through the central QPC in the currently interested Andreev-like tunneling limit.
- In the latest version of Fig. 1, we have added a new subfigure (panel d) to illustrate the generation of non-equilibrium anyons, together with its physical significance. Corresponding discussions are also introduced in the main text.
- We have added a new figure, Fig. 2, to illustrate the newly introduced braiding process, i.e., the anyon-quasihole braiding. This process is further compared to time-domain braiding, another focused braiding process, in the SI Sec. II and SI Fig. S3. This comparison involves both mathematical expressions and physical interpretations.
- We stress the crucial role of particle collisions in the generation of entanglement in our system, in Sec. 2 and Fig. 3 of the latest main text. In addition, we have changed I_T^{double} and S_T^{double} into $I_T^{\text{collision}}$ and $S_T^{\text{collision}}$, to stress the significance of two-particle collision within our work.
- After Eq. (4) of the main text, we have added several lines to stress the missing of $\mathcal{P}_{\text{Andreev}}$ for the fermionic system, $\nu = 1$. In the next paragraph after Eq. (4), we have also stated the influence of interaction on $\mathcal{P}_{\text{Andreev}}$.
- In the caption of the latest Fig. 4, we refer to the values of transmission probabilities \mathcal{T}_A and \mathcal{T}_B of two diluters, and \mathcal{T}_C through the central collider.

- In Sec. 6 *Conclusion* of the latest main text, we have added several lines to discuss the influence of interactions, especially in real experimental setups.
- We have simplified the *Methods*, to make it more accessible to non-expert. Here we have also added Fig. 5, to compare transmissions through the central collider, between the single-source and double-source situations.
- In the latest main text, we have removed most jargons, e.g., “disconnected diagrams”, “time integrals” and “self-contracted”. They now only appear in technical sections of the Supplementary Information. In addition, comparisons to related previous literatures are also moved from the main text, into the SI Sec. II.
- In the latest SI Secs. IB and IC, we provide very detailed derivations on the evaluation of correlation functions to higher orders of dilution. A new figure (SI Fig. S1) of the latest version is further added to illustrate two major contributions.
- We have greatly modified the SI Secs. VB and VC, to discuss the influence of different types of interactions. Specifically, in Sec. VC, we explicitly state that the entanglement pointer becomes advantageous when facing intra-edge interactions, among channels that belong to the same physical boundary.
- We have added the following references:

[55] Samuelsson, P., Sukhorukov, E.V., Büttiker, M.: Two-particle Aharonov-Bohm effect and entanglement in the electronic Hanbury Brown-Twiss setup, *Phys. Rev. Lett.* **92**, 026805 (2004);

[56] Roychowdhury, K., Wadhawan, D., Mehta, P., Karmakar, B., Das, S.: Quantum Hall realization of polarized intensity interferometry, *Phys. Rev. B* **93**, 220101 (2016);

[66] Kitaev, A., Preskill, J.: Topological entanglement entropy, *Phys. Rev. Lett.* **96**, 110404 (2006);

[67] Jonckheere, T., Devillard, P., Crepieux, A., Martin, T.: Electronic mach-zehnder interferometer in the fractional quantum hall effect. *Phys. Rev. B* **72**, 201305 (2005);

[68] Guyon, R., Devillard, P., Martin, T., Safi, I.: Klein factors in multiple fractional quantum hall edge tunneling. *Phys. Rev. B* **65**, 153304 (2002).

Below, we briefly respond to the minor concern mentioned by Reviewer 1.

My only remaining concern pertains to the calculation of the correlator in Section IC of the Supplementary Information. While the explanation has been significantly improved, I still have some reservations regarding the approximations made with respect to the variable "t", which is integrated over the full range from $-\infty$ to $+\infty$ in the computation of transport quantities. In such a context, it is not rigorous to take limits over this variable.

If we understand correctly, Reviewer 1 is possibly pointing to the Supplementary equations, (S20) and (S22), where we provide integral outcomes in two asymptotic limiting cases, $|t| \gg 1/\nu eV$ and $0 < |t| \ll 1/\nu eV$.

We would like to stress that these two asymptotics are actually obtained from Eqs. (S19) and (S21), where integral outcomes, importantly, cover the entire range of t , i.e., from $-\infty$ to $+\infty$. As we stated below Eq. (S41), the integrals over the full range of t from $-\infty$ to $+\infty$ were evaluated with the correlation functions given by Eqs. (S30) and (S31), using the results of Sec. III A. These correlation functions are expressed in terms of the functions $\zeta_{\pm}(\nu, \nu eVt)$, introduced in Eq. (S24), which cover the full range of times.

Nevertheless, the approximations employed by the authors are consistent with those found in other published works, and I acknowledge that they have presented them with commendable clarity, allowing the reader to follow the derivation in a transparent manner.

We thank the Reviewer for the positive assessment of clarity and transparency of the presentation of our calculations. In fact, the asymptotic expressions of Eqs. (S20) and (S22) are provided for the convenience of the readers. Indeed, most recent works that discussed the topic of time-domain braiding, e.g., Refs. [39-42], mainly focused on correlation functions in the $t \gg 1/\nu eV$ limit, as the Reviewer rightly mentions. It is thus convenient for the readers to compare our asymptotic expressions for the correlation functions induced by the anyon-quasihole braiding processes with the time-domain braiding expressions of the previous literature, as we do in Sec. II of Supplementary Information, see Eqs. (S32) and (S33).

In contrast to the previous literature, our work presents the full expressions for the correlation functions, valid not only in the asymptotic time range that actually indeed dominates the integrals for transport observables. This allowed us to rigorously analyze all the contributions to these integrals in Sec. IIIB of the Supplementary Information and to justify all the approximations used in our calculation. Crucially, in the strongly dilute limit, the time integrals are determined by the time range $|t| \sim \nu e/I_{A0,B0} \gg 1/\nu eV$, where the functions $\zeta_{\pm}(\nu, \nu eVt)$ can be replaced as described in Eq. (S38).

Taking all this into account, I recommend the manuscript for publication in Nature Communications without further modifications.

We are happy to learn that Reviewer 1 recommends our manuscript for publication in Nature Communications without any further modifications.